# Proximal Mean Field Learning in Shallow Neural Networks

**Alexis M.H. Teter**                                             *amteter@ucsc.edu*
*Department of Applied Mathematics*
*University of California, Santa Cruz*

**Iman Nodozi**                                                  *inodozi@ucsc.edu*
*Department of Electrical and Computer Engineering*
*University of California, Santa Cruz*

**Abhishek Halder**                                             *ahalder@iastate.edu*
*Department of Aerospace Engineering*
*Iowa State University*

**Reviewed on OpenReview:** *https://openreview.net/forum?id=vyRBsqj5iG*

## Abstract

We propose a custom learning algorithm for shallow over-parameterized neural networks, i.e., networks with single hidden layer having infinite width. The infinite width of the hidden layer serves as an abstraction for the over-parameterization. Building on the recent mean field interpretations of learning dynamics in shallow neural networks, we realize mean field learning as a computational algorithm, rather than as an analytical tool. Specifically, we design a Sinkhorn regularized proximal algorithm to approximate the distributional flow for the learning dynamics over weighted point clouds. In this setting, a contractive fixed point recursion computes the time-varying weights, numerically realizing the interacting Wasserstein gradient flow of the parameter distribution supported over the neuronal ensemble. An appealing aspect of the proposed algorithm is that the measure-valued recursions allow meshless computation. We demonstrate the proposed computational framework of interacting weighted particle evolution on binary and multi-class classification. Our algorithm performs gradient descent of the free energy associated with the risk functional.

## 1 Introduction

While universal function approximation theorems for neural networks have long been known (Cybenko, 1989; Barron, 1993; Hornik et al., 1989), such guarantees do not account for the dynamics of the learning algorithms. Starting in 2018, several works (Mei et al., 2018; Chizat & Bach, 2018; Rotskoff & Vanden-Eijnden, 2018; Sirignano & Spiliopoulos, 2020; Rotskoff & Vanden-Eijnden, 2022; Boursier et al., 2022) pointed out that the first order learning dynamics for shallow (i.e., single hidden layer) neural networks in the infinite width (i.e., over-parameterization) limit leads to a nonlinear partial differential equation (PDE) that depends on a pair of advection and interaction potentials.

The Cauchy initial value problem associated with the PDE describes the evolution of neuronal parameter ensemble induced by the learning dynamics. This result can be interpreted as a dynamical version of the universal approximation theorem. In particular, the potentials depend on both the loss function as well as the activation functions of the neural network.

The advection potential in this nonlinear PDE induces a drift, while the interaction potential induces a nonlocal force. Remarkably, this PDE can be interpreted as an infinite dimensional gradient flow of the population risk w.r.t. the Wasserstein metric arising from the theory of optimal mass transport (Villani, 2009; 2021).

The nonlocal nonlinear PDE interpretation makes connection with the so-called "propagation of chaos"–a term due to Kac (Kac, 1956) that has grown into a substantial literature in statistical physics (McKean Jr, 1966; Sznitman, 1991; Carmona & Delarue, 2018). From this viewpoint, the first order algorithmic dynamics makes the individual neurons in the hidden layer behave as interacting particles. These particle-level or microscopic interactions manifest as a population-level or macroscopic gradient flow.

As an analytic tool, the mean field Wasserstein gradient flow interpretation helps shed light on the convergence of first order learning dynamics in over-parameterized networks. In this work, we propose Wasserstein proximal recursions to realize the mean field learning dynamics as a meshless computational algorithm.

## 1.1 Computational challenges

Transcribing the mean field Wasserstein gradient flow PDE from an analytical tool to a computational algorithm is particularly challenging in the neural network context. This is because the derivation of the PDE in (Mei et al., 2018; Chizat & Bach, 2018; Rotskoff & Vanden-Eijnden, 2018; Sirignano & Spiliopoulos, 2020), and the corresponding infinite dimensional gradient descent interpretation, is an asymptotic consistency result. Specifically, the PDE describes the time evolution of the joint population density (or population measure in general) for the hidden layer neuronal ensemble. By definition, this interpretation is valid in the mean field (infinite width) limit of the single hidden layer. In other words, to leverage the gradient flow PDE perspective in computation, the number of neurons in the hidden layer must be large.

However, from a numerical perspective, explicitly evolving the joint population in the large width regime is problematic. This is because the large width implies that the time-varying joint neuronal population distributions have high dimensional supports. Even though existing software tools routinely deploy stochastic gradient descent (SGD) algorithms at the *microscopic* (i.e., particle) level, it is practically infeasible to estimate the time-varying population distributions using Monte Carlo or other *a posteriori* function approximation algorithms near this limit. One also cannot resort to standard finite difference-type discretization approach for solving this gradient flow PDE because the large width limit brings the curse of dimensionality. Therefore, it is not obvious whether the mean field dynamics can lead to a learning algorithm in practice.

## 1.2 Related works

Beyond mean field learning, Wasserstein gradient flows appear in many other scientific (Ambrosio et al., 2005; Santambrogio, 2017) and engineering (Halder & Georgiou, 2017; Caluya & Halder, 2021b; Halder et al., 2022) contexts. Thus, there exists a substantial literature on numerically implementing the Wasserstein gradient flows – both with grid (Peyré, 2015; Benamou et al., 2016; Carlier et al., 2017; Carrillo et al., 2022) and without grid (Liu et al., 2019; Caluya & Halder, 2019a; Halder et al., 2020). The latter class of algorithms are more relevant for the mean field learning context since the underlying parameter space (i.e., the support) is high dimensional. The proximal recursion we consider is closely related to the forward or backward discretization (Salim et al., 2020; Frogner & Poggio, 2020) of the Jordan-Kinderlehrer-Otto (JKO) scheme (Jordan et al., 1998).

To bypass numerical optimization over the manifold of measures or densities, recent works (Mokrov et al., 2021; Alvarez-Melis et al., 2021; Bunne et al., 2022) propose using input convex neural networks (Amos et al., 2017) to perform the Wasserstein proximal time-stepping by learning the convex potentials (Brenier, 1991) associated with the respective pushforward maps. To alleviate the computational difficulties in high dimensions, Bonet et al. (2021) proposes replacing the Wasserstein distance with the sliced-Wasserstein distance (Rabin et al., 2011) scaled by the ambient dimension.

We note here that there exists extensive literature on the mean field limit of learning in neural networks from other perspectives, including the Neural Tangent Kernel (NTK) and the Gaussian process viewpoints, see e.g., (Jacot et al., 2018; Lee et al., 2019; Novak et al., 2019; Xiao et al., 2018; Li & Nguyen, 2018; Matthews et al., 2018). Roughly speaking, the key observation is that in the infinite width limit, the learning evolves as a suitably defined Gaussian process with network architecture-dependent kernel. We mention this in the passing since in this work, we will only focus on the Wasserstein gradient flow viewpoint. We point out that

unlike the NTK, the mean field limit in Wasserstein gradient flow viewpoint does not approximate dynamical nonlinearity. In particular, the associated initial value problem involves a nonlinear PDE (see (14)).

### 1.3 Contributions

With respect to the related works referenced above, the main contribution of the present work is that we propose a meshless Wasserstein proximal algorithm that directly implements the macroscopic learning dynamics in a fully connected shallow network. We do so by evolving population densities as *weighted* scattered particles.

Different from Monte Carlo-type algorithms, the weight updates in our setting are done explicitly by solving a regularized dual ascent. This computation occurs within the dashed box highlighted in Fig. 1. The particles' location updates are done via *nonlocal* Euler-Maruyama. These two updates interact with each other (see Fig. 1), and together set up a discrete time-stepping scheme.

The discrete time-stepping procedure we propose, is a novel interacting particle system in the form of a meshless algorithm. Our contribution advances the state-of-the-art as it allows for evolving the neuronal population distribution in an online manner, as needed in mean field learning. This is in contrast to *a posteriori* function approximation in existing Monte Carlo methods (cf. Sec. 1.1). Explicit proximal weight updates allows us to bypass offline high dimensional function approximation, thereby realizing mean field learning at an algorithm level.

With respect to the computational challenges mentioned in Sec. 1.1, it is perhaps surprising that we are able to design an algorithm for explicitly evolving the population densities without directly discretizing the spatial domain of the underlying PDE. Our main idea to circumvent the computational difficulty is to solve the gradient flow PDE *not* as a PDE, but to instead direct the algorithmic development for a proximal recursion associated with the gradient flow PDE. This allows us to implement the associated proximal recursion over a suitably discrete time without directly discretizing the parameter space (the latter is what makes the computation otherwise problematic in the mean field regime).

For specificity, we illustrate the proposed framework on two numerical experiments involving binary and multi-class classification with quadratic risk. The proposed methodology should be of broad interest to other problems such as the policy optimization (Zhang et al., 2018; Chu et al., 2019; Zhang et al., 2020) and the adversarial learning (Domingo-Enrich et al., 2020; Mroueh & Nguyen, 2021; Lu, 2023).

We emphasize that the perspective taken in this work is somewhat non-standard w.r.t. the existing literature in that our main intent is to explore the possibility of designing a new class of algorithms by leveraging the connection between the mean field PDE and the Wasserstein proximal operator. This is a new line of idea for learning algorithm design that we show is feasible. As such, we do not aim to immediately surpass the carefully engineered existing state-of-the-art in experiments. Instead, this study demonstrates a proof-of-concept which should inspire follow up works.

### 1.4 Notations and preliminaries

We use the standard convention where the boldfaced lowercase letters denote vectors, boldfaced uppercase letters denote matrices, and non-boldfaced letters denote scalars. We use the symbols $\nabla$ and $\Delta$ to denote the Euclidean gradient and Laplacian operators, respectively. In case of potential confusion, we attach subscripts to these symbols to clarify the differential operator is taken w.r.t. which variable. The symbols $\odot, \oslash$, exp, and tanh denote *elementwise* multiplication, division, exponential, and hyperbolic tangent, respectively. Furthermore, rand and randn denote draw of the uniform and standard normal distributed random vector of appropriate dimension. In addition, $\mathbf{1}$ represents a vector of ones of appropriate dimension.

Let $\mathcal{Z}_1, \mathcal{Z}_2 \subseteq \mathbb{R}^d$. The squared 2-Wasserstein metric $W_2$ (with standard Euclidean ground cost) between two probability measures $\pi_1(\mathrm{d}\boldsymbol{z}_1)$ and $\pi_2(\mathrm{d}\boldsymbol{z}_2)$ (or between the corresponding densities when the measures are

absolutely continuous), where $\boldsymbol{z}_1 \in \mathcal{Z}_1$, $\boldsymbol{z}_2 \in \mathcal{Z}_2$, is defined as

$$W_2^2\left(\pi_1, \pi_2\right) := \inf_{\pi \in \Pi(\pi_1, \pi_2)} \int_{\mathcal{Z}_1 \times \mathcal{Z}_2} \|\boldsymbol{z}_1 - \boldsymbol{z}_2\|_2^2 \,\mathrm{d}\pi(\boldsymbol{z}_1, \boldsymbol{z}_2). \tag{1}$$

In (1), the symbol $\Pi\left(\pi_1, \pi_2\right)$ denotes the collection of joint measures (couplings) with finite second moments, whose first marginal is $\pi_1$, and the second marginal is $\pi_2$.

## 1.5 Organization

The remainder of this paper is organized as follows. In Sec. 2, we provide the necessary background for the empirical risk minimization and for the corresponding mean field limit. The proposed proximal algorithm (including its derivation, convergence guarantee and implementation) is detailed in Sec. 3. We then report numerical case studies in Sec. 4 and Sec. 5 for binary and multi-class classifications, respectively. Sec. 6 concludes the paper. Supporting proofs and derivations are provided in Appendices A, B and C. Numerical results for a synthetic one dimensional case study of learning a sinusoid using the proposed proximal algorithm is provided in Appendix D.

## 2 From empirical risk minimization to proximal mean field learning

To motivate the mean field learning formulation, we start by discussing the more familiar empirical risk minimization set up. We then explain the infinite width limit for the same.

### 2.1 Empirical risk minimization

We consider a supervised learning problem where the dataset comprises of the features $\boldsymbol{x} \in \mathcal{X} \subseteq \mathbb{R}^{n_x}$, and the labels $y \in \mathcal{Y} \subseteq \mathbb{R}$, i.e., the samples of the dataset are tuples of the form

$$(\boldsymbol{x}, y) \in \mathcal{X} \times \mathcal{Y} \subseteq \mathbb{R}^{n_x} \times \mathbb{R}.$$

The objective of the supervised learning problem is to find the parameter vector $\boldsymbol{\theta} \in \mathbb{R}^p$ such that $y \approx f(\boldsymbol{x}, \boldsymbol{\theta})$ where $f$ is some function class parameterized by $\boldsymbol{\theta}$. In other words, $f$ maps from the feature space $\mathcal{X}$ to the label space $\mathcal{Y}$. To this end, we consider a shallow neural network with a single hidden layer having $n_\mathrm{H}$ neurons. Then, the parameterized function $f$ admits representation

$$f(\boldsymbol{x}, \boldsymbol{\theta}) := \frac{1}{n_\mathrm{H}} \sum_{i=1}^{n_\mathrm{H}} \Phi\left(\boldsymbol{x}, \boldsymbol{\theta}_i\right), \tag{2}$$

where $\Phi\left(\boldsymbol{x}, \boldsymbol{\theta}_i\right) := a_i \sigma(\langle \boldsymbol{w}_i, \boldsymbol{x}\rangle + b_i)$ for all $i \in [n_\mathrm{H}] := \{1, 2, \ldots, n_\mathrm{H}\}$, and $\sigma(\cdot)$ is a smooth activation function. The parameters $a_i, \boldsymbol{w}_i$ and $b_i$ are the scaling, weights, and bias of the $i^\mathrm{th}$ hidden neuron, respectively, and together comprise the specific parameter realization $\boldsymbol{\theta}_i \in \mathbb{R}^p$, $i \in [n_\mathrm{H}]$.

We stack the parameter vectors of all hidden neurons as

$$\overline{\boldsymbol{\theta}} := \left(\boldsymbol{\theta}_1, \boldsymbol{\theta}_2, \ldots, \boldsymbol{\theta}_{n_\mathrm{H}}\right)^\top \in \mathbb{R}^{p n_\mathrm{H}}$$

and consider minimizing the following quadratic loss:

$$l(y, \boldsymbol{x}, \overline{\boldsymbol{\theta}}) \equiv l(y, f(\boldsymbol{x}, \overline{\boldsymbol{\theta}})) := \underbrace{\left(y - f(\boldsymbol{x}, \overline{\boldsymbol{\theta}})\right)^2}_{\text{quadratic loss}}. \tag{3}$$

We suppose that the training data follows the joint probability distribution $\gamma$, i.e., $(\boldsymbol{x}, y) \sim \gamma$. Define the *population risk* $R$ as the expected loss given by

$$R(f) := \mathbb{E}_{(\boldsymbol{x}, y) \sim \gamma}[l(y, \boldsymbol{x}, \overline{\boldsymbol{\theta}})]. \tag{4}$$

In practice, $\gamma$ is unknown, so we approximate the population risk with the *empirical risk*

$$R(f) \approx \frac{1}{n_{\text{data}}} \sum_{j=1}^{n_{\text{data}}} l\left(y_j, \boldsymbol{x}_j, \bar{\boldsymbol{\theta}}\right) \tag{5}$$

where $n_{\text{data}}$ is the number of data samples. Then, the supervised learning problem reduces to the empirical risk minimization problem

$$\min_{\bar{\boldsymbol{\theta}} \in \mathbb{R}^{pn_{\text{H}}}} R(f). \tag{6}$$

Problem (6) is a large but finite dimensional optimization problem that is nonconvex in the decision variable $\bar{\boldsymbol{\theta}}$. The standard approach is to employ first or second order search algorithms such as the variants of SGD or ADAM (Kingma & Ba, 2014).

## 2.2 Mean field limit

The mean field limit concerns with a continuum of hidden layer neuronal population by letting $n_{\text{H}} \to \infty$. Then, we view (2) as the empirical average associated with the ensemble average

$$f_{\text{MeanField}} := \int_{\mathbb{R}^p} \Phi(\boldsymbol{x}, \boldsymbol{\theta}) \underbrace{\mathrm{d}\mu(\boldsymbol{\theta})}_{\text{hidden neuronal population mass}} = \mathbb{E}_{\boldsymbol{\theta}}[\Phi(\boldsymbol{x}, \boldsymbol{\theta})], \tag{7}$$

where $\mu$ denotes the joint population measure supported on the hidden neuronal parameter space in $\mathbb{R}^p$. Assuming the absolute continuity of $\mu$ for all times, we write $\mathrm{d}\mu(\boldsymbol{\theta}) = \rho(\boldsymbol{\theta})\mathrm{d}\boldsymbol{\theta}$ where $\rho$ denotes the joint population density function (PDF).

Thus, the risk functional $R$, now viewed as a function of the joint PDF $\rho$, takes the form

$$F(\rho) := R(f_{\text{MeanField}}(\boldsymbol{x}, \rho)) = \mathbb{E}_{(\boldsymbol{x}, y)}\left(y - \int_{\mathbb{R}^p} \Phi(\boldsymbol{x}, \boldsymbol{\theta})\rho(\boldsymbol{\theta})\mathrm{d}\boldsymbol{\theta}\right)^2$$

$$= F_0 + \int_{\mathbb{R}^p} V(\boldsymbol{\theta})\rho(\boldsymbol{\theta})\mathrm{d}\boldsymbol{\theta} + \int_{\mathbb{R}^{2p}} U(\boldsymbol{\theta}, \tilde{\boldsymbol{\theta}})\rho(\boldsymbol{\theta})\rho(\tilde{\boldsymbol{\theta}})\mathrm{d}\boldsymbol{\theta}\mathrm{d}\tilde{\boldsymbol{\theta}}, \tag{8}$$

where

$$F_0 := \mathbb{E}_{(\boldsymbol{x}, y)}\left[y^2\right], \quad V(\boldsymbol{\theta}) := \mathbb{E}_{(\boldsymbol{x}, y)}[-2y\Phi(\boldsymbol{x}, \boldsymbol{\theta})], \quad U(\boldsymbol{\theta}, \tilde{\boldsymbol{\theta}}) := \mathbb{E}_{(\boldsymbol{x}, y)}[\Phi(\boldsymbol{x}, \boldsymbol{\theta})\Phi(\boldsymbol{x}, \tilde{\boldsymbol{\theta}})]. \tag{9}$$

Therefore, the supervised learning problem, in this mean field limit, becomes an infinite dimensional variational problem:

$$\min_{\rho} F(\rho) \tag{10}$$

where $F$ is a sum of three functionals. The first summand $F_0$ is independent of $\rho$. The second summand is a potential energy given by expected value of "drift potential" $V$ and is linear in $\rho$. The last summand is a bilinear interaction energy involving an "interaction potential" $U$ and is nonlinear in $\rho$.

The main result in Mei et al. (2018) was that using first order SGD learning dynamics induces a gradient flow of the functional $F$ w.r.t. the 2-Wasserstein metric $W_2$, i.e., the mean field learning dynamics results in a joint PDF trajectory $\rho(t, \boldsymbol{\theta})$. Then, the minimizer in (10) can be obtained from the large $t$ limit of the following nonlinear PDE:

$$\frac{\partial \rho}{\partial t} = -\nabla^{W_2} F(\rho), \tag{11}$$

where the 2-Wasserstein gradient (Villani, 2021, Ch. 9.1) (Ambrosio et al., 2005, Ch. 8) of $F$ is

$$\nabla^{W_2} F(\rho) := -\nabla \cdot \left(\rho \nabla \frac{\delta F}{\partial \rho}\right),$$

and $\frac{\delta}{\delta \rho}$ denotes the functional derivative w.r.t. $\rho$.

In particular, Mei et al. (2018) considered the regularized risk functional[1]

$$F_\beta(\rho) := F(\rho) + \beta^{-1} \int_{\mathbb{R}^p} \rho \log \rho \, \mathrm{d}\boldsymbol{\theta}, \quad \beta > 0, \tag{12}$$

by adding a strictly convex regularizer (scaled negative entropy) to the unregularized risk $F$. In that case, the sample path dynamics corresponding to the macroscopic dynamics (11) precisely becomes the noisy SGD:

$$\mathrm{d}\boldsymbol{\theta} = -\nabla_{\boldsymbol{\theta}} \left( V(\boldsymbol{\theta}) + \int_{\mathbb{R}^p} U(\boldsymbol{\theta}, \tilde{\boldsymbol{\theta}}) \rho(\tilde{\boldsymbol{\theta}}) \mathrm{d}\tilde{\boldsymbol{\theta}} \right) \mathrm{d}t + \sqrt{2\beta^{-1}} \, \mathrm{d}\boldsymbol{\eta}, \quad \boldsymbol{\theta}(t=0) \sim \rho_0, \tag{13}$$

where $\boldsymbol{\eta}$ is the standard Wiener process in $\mathbb{R}^p$, and the random initial condition $\boldsymbol{\theta}(t=0)$ follows the law of a suitable PDF $\rho_0$ supported over $\mathbb{R}^p$.

In this regularized case, (11) results in the following nonlinear PDE initial value problem (IVP):

$$\frac{\partial \rho}{\partial t} = \nabla_{\boldsymbol{\theta}} \cdot \left( \rho \left( V(\boldsymbol{\theta}) + \int_{\mathbb{R}^p} U(\boldsymbol{\theta}, \tilde{\boldsymbol{\theta}}) \rho(\tilde{\boldsymbol{\theta}}) \mathrm{d}\tilde{\boldsymbol{\theta}} \right) \right) + \beta^{-1} \Delta_{\boldsymbol{\theta}} \rho, \quad \rho(t=0, \boldsymbol{\theta}) = \rho_0. \tag{14}$$

In other words, the noisy SGD induces evolution of a PDF-valued trajectory governed by the advection, nonlocal interaction, and diffusion–the latter originating from regularization. Notice that a large value of $\beta > 0$ implies a small entropic regularization in (12), hence a small additive process noise in (13), and consequently, a small diffusion term in the PDE (14).

The regularized risk functional $F_\beta$ in (12) can be interpreted as a free energy wherein $F$ contributes a sum of the advection potential energy and interaction energy. The term $\beta^{-1} \int_{\mathbb{R}^p} \rho \log \rho \, \mathrm{d}\boldsymbol{\theta}$ contributes an internal energy due to the noisy fluctuations induced by the additive Wiener process noise $\sqrt{2\beta^{-1}} \mathrm{d}\boldsymbol{\eta}$ in (13).

In Mei et al. (2018), asymptotic guarantees were obtained for the solution of (14) to converge to the minimizer of $F_\beta$. Our idea, outlined next, is to solve the minimization of $F_\beta$ using measure-valued proximal recursions.

## 2.3 Proximal mean field learning

For numerically computing the solution of the PDE IVP (14), we propose proximal recursions over $\mathcal{P}_2(\mathbb{R}^p)$, defined as the manifold of joint PDFs supported over $\mathbb{R}^p$ having finite second moments. Symbolically,

$$\mathcal{P}_2(\mathbb{R}^p) := \left\{ \text{Lebesgue integrable } \rho \text{ over } \mathbb{R}^p \mid \rho \geq 0, \int_{\mathbb{R}^p} \rho \, \mathrm{d}\boldsymbol{\theta} = 1, \int_{\mathbb{R}^p} \boldsymbol{\theta}^\top \boldsymbol{\theta} \rho \, \mathrm{d}\boldsymbol{\theta} < \infty \right\}.$$

Proximal updates generalize the concept of gradient steps, and are of significant interest in both finite and infinite dimensional optimization (Rockafellar, 1976a;b; Bauschke et al., 2011; Teboulle, 1992; Bertsekas et al., 2011; Parikh et al., 2014). For a given input, these updates take the form of a structured optimization problem:

proximal update

$$= \underset{\text{decision variable}}{\arg\inf} \left\{ \frac{1}{2} \mathrm{dist}^2 \left( \text{decision variable}, \text{input} \right) + \text{time step} \times \text{functional} \left( \text{decision variable} \right) \right\}, \tag{15}$$

for some suitable notion of distance $\mathrm{dist}(\cdot, \cdot)$ on the space of decision variables, and some associated functional. It is usual to view (15) as an *operator* mapping input $\mapsto$ proximal update, thus motivating the term *proximal operator*.

The connection between (15) and the gradient flow comes from recursively evaluating (15) with some initial choice for the input. For suitably designed pair $(\mathrm{dist}(\cdot, \cdot), \text{functional})$, in the small time step limit, the sequence of proximal updates generated by (15) converge to the infimum of the functional. In other words, the gradient descent of the functional w.r.t. dist may be computed as the fixed point of the proximal operator

---

[1]The parameter $\beta > 0$ is referred to as the inverse temperature.

(15). For a parallel between gradient descent and proximal recursions in finite and infinite dimensional gradient descent, see e.g., (Halder & Georgiou, 2017, Sec. I). Infinite dimensional proximal recursions over the manifold of PDFs have recently appeared in uncertainty propagation (Caluya & Halder, 2019b; Halder et al., 2022), stochastic filtering (Halder & Georgiou, 2017; 2018; 2019), and stochastic optimal control (Caluya & Halder, 2021b;a).

In our context, the decision variable $\rho \in \mathcal{P}_2(\mathbb{R}^p)$ and the distance metric dist $\equiv W_2$. Specifically, we propose recursions over discrete time $t_{k-1} := (k-1)h$ where the index $k \in \mathbb{N}$, and $h > 0$ is a constant time stepsize. Leveraging that (14) describes gradient flow of the functional $F_\beta$ w.r.t. the $W_2$ distance metric, the associated proximal recursion is of the form

$$\varrho_k = \text{prox}_{hF_\beta}^{W_2}(\varrho_{k-1}) := \underset{\varrho \in \mathcal{P}_2(\mathbb{R}^p)}{\arg\inf} \left\{ \frac{1}{2}\left(W_2\left(\varrho, \varrho_{k-1}\right)\right)^2 + h\, F_\beta\left(\varrho\right) \right\} \tag{16}$$

where $\varrho_{k-1}(\cdot) := \varrho(\cdot, t_{k-1})$, and $\varrho_0 \equiv \rho_0$. The notation $\text{prox}_{hF_\beta}^{W_2}(\varrho_{k-1})$ can be parsed as "the proximal operator of the scaled functional $hF_\beta$ w.r.t. the distance $W_2$, acting on the input $\varrho_{k-1} \in \mathcal{P}_2(\mathbb{R}^p)$". Our idea is to evaluate the recursion in the small $h$ limit, i.e., for $h \downarrow 0$.

To account for the nonconvex bilinear term appearing in (12), following (Benamou et al., 2016, Sec. 4), we employ the approximation:

$$\int_{\mathbb{R}^{2p}} U(\boldsymbol{\theta}, \tilde{\boldsymbol{\theta}})\varrho(\boldsymbol{\theta})\varrho(\tilde{\boldsymbol{\theta}})\mathrm{d}\boldsymbol{\theta}\mathrm{d}\tilde{\boldsymbol{\theta}} \approx \int_{\mathbb{R}^{2p}} U(\boldsymbol{\theta}, \tilde{\boldsymbol{\theta}})\varrho(\boldsymbol{\theta})\varrho_{k-1}(\tilde{\boldsymbol{\theta}})\mathrm{d}\boldsymbol{\theta}\mathrm{d}\tilde{\boldsymbol{\theta}} \quad \forall k \in \mathbb{N}.$$

We refer to the resulting approximation of $F_\beta$ as $\hat{F}_\beta$, i.e.,

$$\hat{F}_\beta\left(\varrho, \varrho_{k-1}\right) := \int_{\mathbb{R}^p} \left( F_0 + V(\boldsymbol{\theta}) + \left( \int_{\mathbb{R}^p} U(\boldsymbol{\theta}, \tilde{\boldsymbol{\theta}})\varrho_{k-1}(\tilde{\boldsymbol{\theta}})\mathrm{d}\tilde{\boldsymbol{\theta}} \right) + \beta^{-1}\log\varrho(\boldsymbol{\theta}) \right)\varrho(\boldsymbol{\theta})\mathrm{d}\boldsymbol{\theta}.$$

Notice in particular that $\hat{F}_\beta$ depends on both $\varrho, \varrho_{k-1}$, $k \in \mathbb{N}$. Consequently, this approximation results in a *semi-implicit* variant of (16), given by

$$\varrho_k := \underset{\varrho \in \mathcal{P}_2(\mathbb{R}^p)}{\arg\inf} \left\{ \frac{1}{2}\left(W_2\left(\varrho, \varrho_{k-1}\right)\right)^2 + h\hat{F}_\beta\left(\varrho, \varrho_{k-1}\right) \right\}. \tag{17}$$

We have the following consistency guarantee, stated informally, among the solution of the PDE IVP (14) and that of the variational recursions (17). The rigorous statement and proof are provided in Appendix A.

**Theorem 1.** *(Informal) Consider the regularized risk functional (12) wherein $F$ is given by (8)-(9). In the small time step ($h \downarrow 0$) limit, the proximal updates (17) with $\varrho_0 \equiv \rho_0$ converge to the solution for the PDE IVP (14).*

We next detail the proposed algorithmic approach to numerically solve (17).

## 3 ProxLearn: proposed proximal algorithm

The overall workflow of our proposed proximal mean field learning framework is shown in Fig. 1. We generate $N$ samples from the known initial joint PDF $\varrho_0$ and store them as a weighted point cloud $\{\boldsymbol{\theta}_0^i, \varrho_0^i\}_{i=1}^N$. Here, $\varrho_0^i := \varrho_0(\boldsymbol{\theta}_0^i)$ for all $i \in [N]$. In other words, the weights of the samples are the joint PDF values evaluated at those samples.

For each $k \in \mathbb{N}$, the weighted point clouds $\{\boldsymbol{\theta}_k^i, \varrho_k^i\}_{i=1}^N$ are updated through the two-step process outlined in our proposed Algorithm 1, referred to as PROXLEARN. At a high level, lines 9–18 in Algorithm 1 perform nonlinear block co-ordinate recursion on internally defined vectors $\boldsymbol{z}, \boldsymbol{q}$ whose converged values yield the proximal update (line 19). We next explain where these recursions come from detailing both the derivation of PROXLEARN and its convergence guarantee.

### 3.1 Derivation of ProxLearn

The main idea behind our derivation that follows, is to regularize and dualize the discrete version of the optimization problem in (17). This allows us to leverage certain structure of the optimal solution that emerges from the first order conditions for optimality, which in turn helps design a custom numerical recursion.

Specifically, to derive the recursion given in PROXLEARN, we first write the discrete version of (17) as

$$\boldsymbol{\varrho}_k = \arg\min_{\boldsymbol{\varrho}} \left\{ \min_{\boldsymbol{M} \in \Pi(\boldsymbol{\varrho}_{k-1}, \boldsymbol{\varrho})} \frac{1}{2} \langle \boldsymbol{C}_k, \boldsymbol{M} \rangle + h \left\langle \boldsymbol{v}_{k-1} + \boldsymbol{U}_{k-1} \boldsymbol{\varrho}_{k-1} + \beta^{-1} \log \boldsymbol{\varrho}, \boldsymbol{\varrho} \right\rangle \right\}, \quad k \in \mathbb{N}, \tag{18}$$

where

$$\Pi(\boldsymbol{\varrho}_{k-1}, \boldsymbol{\varrho}) := \{ \boldsymbol{M} \in \mathbb{R}^{N \times N} \mid \boldsymbol{M} \geq \boldsymbol{0} \text{ (elementwise)}, \boldsymbol{M}\boldsymbol{1} = \boldsymbol{\varrho}_{k-1}, \boldsymbol{M}^\top \boldsymbol{1} = \boldsymbol{\varrho} \}, \tag{19}$$

$$\boldsymbol{v}_{k-1} \equiv V(\boldsymbol{\theta}_{k-1}), \tag{20}$$

$$\boldsymbol{U}_{k-1} \equiv U\left(\boldsymbol{\theta}_{k-1}, \tilde{\boldsymbol{\theta}}_{k-1}\right), \tag{21}$$

and $\boldsymbol{C}_k \in \mathbb{R}^{N \times N}$ denotes the squared Euclidean distance matrix, i.e.,

$$\boldsymbol{C}_k(i, j) := \|\boldsymbol{\theta}_k^i - \boldsymbol{\theta}_{k-1}^j\|_2^2 \quad \forall (i, j) \in [N] \times [N].$$

We next follow a "regularize-then-dualize" approach. In particular, we regularize (18) by adding the entropic regularization $H(\boldsymbol{M}) := \langle \boldsymbol{M}, \log \boldsymbol{M} \rangle$, and write

$$\boldsymbol{\varrho}_k = \arg\min_{\boldsymbol{\varrho}} \left\{ \min_{\boldsymbol{M} \in \Pi(\boldsymbol{\varrho}_{k-1}, \boldsymbol{\varrho})} \frac{1}{2} \langle \boldsymbol{C}_k, \boldsymbol{M} \rangle + \epsilon H(\boldsymbol{M}) + h \left\langle \boldsymbol{v}_{k-1} + \boldsymbol{U}_{k-1} \boldsymbol{\varrho}_{k-1} + \beta^{-1} \log \boldsymbol{\varrho}, \boldsymbol{\varrho} \right\rangle \right\}, \quad k \in \mathbb{N} \tag{22}$$

where $\epsilon > 0$ is a regularization parameter.

Following Karlsson & Ringh (2017, Lemma 3.5), Caluya & Halder (2019a, Sec. III), the Lagrange dual problem associated with (22) is

$$\left(\boldsymbol{\lambda}_0^{\mathrm{opt}}, \boldsymbol{\lambda}_1^{\mathrm{opt}}\right) = \arg\max_{\boldsymbol{\lambda}_0, \boldsymbol{\lambda}_1 \in \mathbb{R}^N} \left\{ \langle \boldsymbol{\lambda}_0, \boldsymbol{\varrho}_{k-1} \rangle - \hat{F}_\beta^\star(-\boldsymbol{\lambda}_1) - \frac{\epsilon}{h} \left( \exp\left(\boldsymbol{\lambda}_0^\top h/\epsilon\right) \exp\left(-\boldsymbol{C}_k/2\epsilon\right) \exp\left(\boldsymbol{\lambda}_1 h/\epsilon\right) \right) \right\} \tag{23}$$

where

$$\hat{F}_\beta^\star(\cdot) := \sup_\vartheta \{ \langle \cdot, \vartheta \rangle - \hat{F}_\beta(\vartheta) \} \tag{24}$$

is the Legendre-Fenchel conjugate of the free energy $\hat{F}_\beta$ in (17), and the optimal coupling matrix $\boldsymbol{M}^{\mathrm{opt}} := [m^{\mathrm{opt}}(i, j)]_{i,j=1}^N$ in (22) has the Sinkhorn form

$$m^{\mathrm{opt}}(i, j) = \exp\left(\boldsymbol{\lambda}_0(i) h/\epsilon\right) \exp\left(-\boldsymbol{C}_k(i, j)/(2\epsilon)\right) \exp\left(\boldsymbol{\lambda}_1(j) h/\epsilon\right). \tag{25}$$

To solve (23), considering (12), we write the "discrete free energy" as

$$\hat{F}_\beta(\boldsymbol{\varrho}) = \left\langle \boldsymbol{v}_{k-1} + \boldsymbol{U}_{k-1} \boldsymbol{\varrho}_{k-1} + \beta^{-1} \log \boldsymbol{\varrho}, \boldsymbol{\varrho} \right\rangle. \tag{26}$$

Its Legendre-Fenchel conjugate, by (24), is

$$\hat{F}_\beta^\star(\boldsymbol{\lambda}) := \sup_\vartheta \{ \boldsymbol{\lambda}^\top \boldsymbol{\varrho} - \boldsymbol{v}_{k-1}^\top \boldsymbol{\varrho} - \boldsymbol{\varrho}^\top \boldsymbol{U}_{k-1} \boldsymbol{\varrho}_{k-1} - \beta^{-1} \boldsymbol{\varrho}^\top \log \boldsymbol{\varrho} \}. \tag{27}$$

Setting the gradient of the objective in (27) w.r.t. $\boldsymbol{\varrho}$ to zero, and solving for $\boldsymbol{\varrho}$ gives the maximizer

$$\boldsymbol{\varrho}_{\max} = \exp\left(\beta \left(\boldsymbol{\lambda} - \boldsymbol{v}_{k-1} - \beta^{-1}\boldsymbol{1} - \boldsymbol{U}_{k-1} \boldsymbol{\varrho}_{k-1}\right)\right). \tag{28}$$

Substituting (28) back into (27), we obtain

$$\hat{F}_\beta^\star(\lambda) = \beta^{-1}\mathbf{1}\exp\left(\beta\left(\lambda - \boldsymbol{v}_{k-1} - \boldsymbol{U}_{k-1}\boldsymbol{\varrho}_{k-1}\right) - \mathbf{1}\right). \tag{29}$$

Fixing $\boldsymbol{\lambda}_0$, and taking the gradient of the objective in (23) w.r.t. $\boldsymbol{\lambda}_1$ gives

$$\exp\left(\boldsymbol{\lambda}_1 h/\epsilon\right)\odot\left(\exp\left(-\boldsymbol{C}_k/2\epsilon\right)^\top\exp\left(\boldsymbol{\lambda}_0 h/\epsilon\right)\right) = \exp(-\beta\boldsymbol{v}_{k-1} - \beta\boldsymbol{U}_{k-1}\boldsymbol{\varrho}_{k-1} - \mathbf{1})\odot\left(\exp\left(\boldsymbol{\lambda}_1 h/\epsilon\right)\right)^{-\frac{\beta\epsilon}{h}}. \tag{30}$$

Likewise, fixing $\boldsymbol{\lambda}_1$, and taking the gradient of the objective in (23) w.r.t. $\boldsymbol{\lambda}_0$, gives

$$\exp\left(\boldsymbol{\lambda}_0 h/\epsilon\right)\odot\left(\exp\left(-\boldsymbol{C}_k/2\epsilon\right)\exp\left(\boldsymbol{\lambda}_1 h/\epsilon\right)\right) = \boldsymbol{\varrho}_{k-1}. \tag{31}$$

Next, letting $\boldsymbol{\Gamma}_k := \exp\left(-\boldsymbol{C}_k/2\epsilon\right)$, $\boldsymbol{q} := \exp\left(\boldsymbol{\lambda}_0 h/\epsilon\right)$, $\boldsymbol{z} := \exp\left(\boldsymbol{\lambda}_1 h/\epsilon\right)$, and $\boldsymbol{\xi}_{k-1} := \exp(-\beta\boldsymbol{v}_{k-1} - \beta\boldsymbol{U}_{k-1}\boldsymbol{\varrho}_{k-1} - \mathbf{1})$, we express (30) as

$$\boldsymbol{z}\odot\left(\boldsymbol{\Gamma}_k^\top\boldsymbol{q}\right) = \boldsymbol{\xi}_{k-1}\odot\boldsymbol{z}^{-\frac{\beta\epsilon}{h}}, \tag{32}$$

and (31) as

$$\boldsymbol{q}\odot\left(\boldsymbol{\Gamma}_k\boldsymbol{z}\right) = \boldsymbol{\varrho}_{k-1}. \tag{33}$$

Finally using (19), we obtain

$$\boldsymbol{\varrho}_k = \left(\boldsymbol{M}^{\text{opt}}\right)^\top\mathbf{1} = \sum_{j=1}^N m^{\text{opt}}(j,i) = \boldsymbol{z}(i)\sum_{j=1}^N\boldsymbol{\Gamma}_k(j,i)\boldsymbol{q}(j) = \boldsymbol{z}\odot\boldsymbol{\Gamma}_k^\top\boldsymbol{q}. \tag{34}$$

In summary, (34) allows us to numerically perform the proximal update.

**Remark 1.** *Note that in Algorithm 1 (i.e., PROXLEARN) presented in Sec. 3, the lines 11, 12, and 19 correspond to (32), (33) and (34), respectively.*

### 3.2 Convergence of ProxLearn

Our next result provides the convergence guarantee for our proposed PROXLEARN algorithm derived in Sec. 3.1.

**Proposition 1.** *The recursions given in lines 7–18 in Algorithm 1 PROXLEARN, converge to a unique fixed point $(\boldsymbol{q}^{\text{opt}}, \boldsymbol{z}^{\text{opt}}) \in \mathbb{R}_{>0}^N \times \mathbb{R}_{>0}^N$. Consequently, the proximal update (34) (i.e., the evaluation at line 19 in Algorithm 1) is unique.*

*Proof.* Notice that the mappings $(\boldsymbol{q}(:,\ell), \boldsymbol{z}(:,\ell)) \mapsto (\boldsymbol{q}(:,\ell+1), \boldsymbol{z}(:,\ell+1))$ given in lines 11 and 12 in Algorithm 1, are cone preserving since these mappings preserve the product orthant $\mathbb{R}_{>0}^N \times \mathbb{R}_{>0}^N$. This is a direct consequence of the definition of $\boldsymbol{q}, \boldsymbol{z}$ in terms of $\boldsymbol{\lambda}_0, \boldsymbol{\lambda_1}$.

Now the idea is to show that the recursions in lines 11 and 12 in Algorithm 1, as composite nonlinear maps, are in fact contractive w.r.t. a suitable metric on this cone. Following Caluya & Halder (2019a, Theorem 3), the $\boldsymbol{z}$ iteration given in line 11 in Algorithm 1, PROXLEARN, for $\ell = 1, 2, \ldots$, is strictly contractive in the Thompson's part metric (Thompson, 1963) and thanks to the Banach contraction mapping theorem, converges to a unique fixed point $\boldsymbol{z}^{\text{opt}} \in \mathbb{R}_{>0}^N$.

We note that our definition of $\boldsymbol{\xi}_{k-1}$ is slightly different compared to the same in Caluya & Halder (2019a, Theorem 3), but this does not affect the proof. From definition of $\boldsymbol{C}_k$, we have $\boldsymbol{C}_k \in [0, \infty)$ which implies $\boldsymbol{\Gamma}_k(i,j) \in (0,1]$. Therefore, $\boldsymbol{\Gamma}_k$ is a positive linear map for each $k \in \mathbb{N}$. Thus, by (linear) Perron-Frobenius theorem, the linear maps $\boldsymbol{\Gamma}_k$ are contractive. Consequently the $\boldsymbol{q}$ iterates also converge to unique fixed point $\boldsymbol{q}^{\text{opt}} \in \mathbb{R}_{>0}^N$.

Since converged pair $(\boldsymbol{q}^{\text{opt}}, \boldsymbol{z}^{\text{opt}}) \in \mathbb{R}_{>0}^N \times \mathbb{R}_{>0}^N$ is unique, so is the proximal update (34), i.e., the evaluation at line 19 in Algorithm 1. $\qquad\square$

We next discuss the implementation details for the proposed PROXLEARN algorithm.

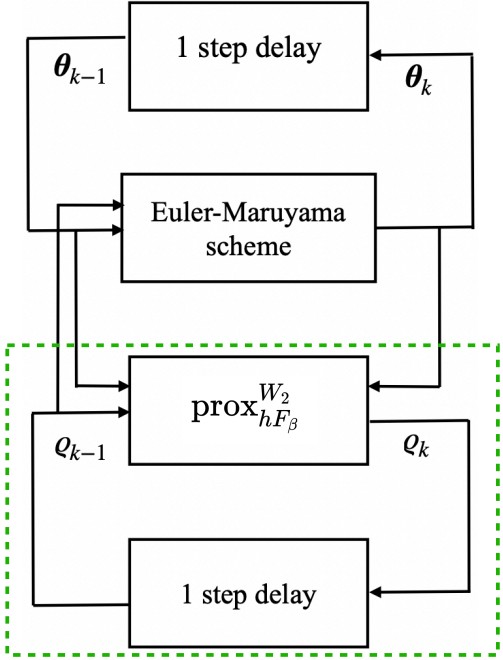

Figure 1: Schematic of the proposed proximal algorithm for mean field learning, updating scattered point cloud $\left\{\boldsymbol{\theta}_{k-1}^i, \varrho_{k-1}^i\right\}_{i=1}^N$ for $k \in \mathbb{N}$. The location of the points $\left\{\boldsymbol{\theta}_{k-1}^i\right\}_{i=1}^N$ are updated via the Euler-Maruyama scheme; the corresponding probability weights are computed via proximal updates highlighted within the dashed box. Explicitly performing the proximal updates via the proposed algorithm, and thereby enabling mean filed learning as an interacting weighted particle system, is our novel contribution

.

### 3.3 Implementation of ProxLearn

We start by emphasizing that PROXLEARN updates both the parameter sample locations $\boldsymbol{\theta}_k^i$ and the joint PDF values $\varrho_k^i$ evaluated at those locations, without gridding the parameter space. In particular, the PDF values are updated online, not as an offline *a posteriori* function approximation as in traditional Monte Carlo algorithms.

We will apply PROXLEARN, as outlined here, in Sec. 4. In Sec. 5, we will detail additional modifications of PROXLEARN to showcase its flexibility.

Required inputs of PROXLEARN are the inverse temperature $\beta$, the time step-size $h$, a regularization parameter $\varepsilon$, and the number of samples $N$. Additional required inputs are the training feature data $\boldsymbol{X} := \left[\boldsymbol{x}_1 \ldots \boldsymbol{x}_{n_{\text{data}}}\right]^\top \in \mathbb{R}^{n_{\text{data}} \times n_x}$ and the corresponding training labels $\boldsymbol{y} := \left[y_1 \ldots y_{n_{\text{data}}}\right]^\top \in \mathbb{R}^{n_{\text{data}}}$, as well the weighted point cloud $\{\boldsymbol{\theta}_{k-1}^i, \varrho_{k-1}^i\}_{i=1}^N$ for each $k \in \mathbb{N}$. Furthermore, PROXLEARN requires two internal parameters as user input: the numerical tolerance $\delta$, and the maximum number of iterations $L$.

For $k \in \mathbb{N}$, let

$$\boldsymbol{\Theta}_{k-1} := \begin{pmatrix} (\boldsymbol{\theta}_{k-1}^1)^\top \\ (\boldsymbol{\theta}_{k-1}^2)^\top \\ \vdots \\ (\boldsymbol{\theta}_{k-1}^N)^\top \end{pmatrix} \in \mathbb{R}^{N \times p}, \quad \boldsymbol{\varrho}_{k-1} := \begin{pmatrix} \varrho_{k-1}^1 \\ \varrho_{k-1}^2 \\ \vdots \\ \varrho_{k-1}^N \end{pmatrix} \in \mathbb{R}_{>0}^N.$$

In line 2 of Algorithm 1, PROXLEARN updates the locations of the parameter vector samples $\boldsymbol{\theta}_k^i$ in $\mathbb{R}^p$ via Algorithm 2, EULERMARUYAMA. This location update takes the form:

$$\boldsymbol{\theta}_k^i = \boldsymbol{\theta}_{k-1}^i - h\nabla \left(V\left(\boldsymbol{\theta}_{k-1}^i\right) + \omega\left(\boldsymbol{\theta}_{k-1}^i\right)\right) + \sqrt{2\beta^{-1}}\left(\boldsymbol{\eta}_k^i - \boldsymbol{\eta}_{k-1}^i\right), \tag{35}$$

---

**Algorithm 1** Proximal Algorithm

---

1: **procedure** PROXLEARN($\boldsymbol{\varrho}_{k-1}, \boldsymbol{\Theta}_{k-1}, \beta, h, \varepsilon, N, \boldsymbol{X}, \boldsymbol{y}, \delta, L$)
2:    $\boldsymbol{v}_{k-1}, \boldsymbol{U}_{k-1}, \boldsymbol{\Theta}_k \leftarrow$ EULERMARUYAMA($h, \beta, \boldsymbol{\Theta}_{k-1}, \boldsymbol{X}, \boldsymbol{y}, \boldsymbol{\varrho}_{k-1}$)  ▷ Update the location of the samples
3:    $\boldsymbol{C}_k(i,j) \leftarrow \left\| \boldsymbol{\theta}_k{}^i - \boldsymbol{\theta}_{k-1}^j \right\|_2^2$
4:    $\boldsymbol{\Gamma}_k \leftarrow \exp(-\boldsymbol{C}_k/2\varepsilon)$                    ▷ Lines 4-8 define the terms needed in re-expressing (30) as (32)
5:    $\boldsymbol{\xi}_{k-1} \leftarrow \exp(-\beta \boldsymbol{v}_{k-1} - \beta \boldsymbol{U}_{k-1}\boldsymbol{\varrho}_{k-1} - \mathbf{1})$
6:    $\boldsymbol{z}_0 \leftarrow \mathrm{rand}_{N \times 1}$
7:    $\boldsymbol{z} \leftarrow [\boldsymbol{z}_0, \mathbf{0}_{N \times (L-1)}]$
8:    $\boldsymbol{q} \leftarrow [\boldsymbol{\varrho}_{k-1} \oslash (\boldsymbol{\Gamma_k z_0}), \mathbf{0}_{N \times (L-1)}]$
9:    $\ell = 1$
10:   **while**  $\ell \leq L$  **do**
11:        $\boldsymbol{z}(:, \ell+1) \leftarrow (\boldsymbol{\xi}_{k-1} \oslash (\boldsymbol{\Gamma}_k^\top \boldsymbol{q}(:, \ell)))^{\frac{1}{1+\beta\varepsilon/h}}$                    ▷ Following (32)
12:        $\boldsymbol{q}(:, \ell+1) \leftarrow \boldsymbol{\varrho}_{k-1} \oslash (\boldsymbol{\Gamma}_k \boldsymbol{z}(:, \ell+1))$                    ▷ Following (33)
13:        **if** $\|\boldsymbol{q}(:, \ell+1) - \boldsymbol{q}(:, \ell)\| < \delta$ and $\|\boldsymbol{z}(:, \ell+1) - \boldsymbol{z}(:, \ell)\| < \delta$ **then**
14:            Break
15:        **else**
16:            $\ell \leftarrow \ell + 1$
17:        **end if**
18:   **end while**
19:   **return** $\boldsymbol{\varrho}_k \leftarrow \boldsymbol{z}(:, \ell) \odot (\boldsymbol{\Gamma}_k^\top \boldsymbol{q}(:, \ell))$                    ▷ Use (34) to map $\boldsymbol{\varrho}_{k-1}$ to $\boldsymbol{\varrho}_k$
20: **end procedure**

---

**Algorithm 2** Euler-Maruyama Algorithm

---

1: **procedure** EULERMARUYAMA($h, \beta, \boldsymbol{\Theta}_{k-1}, \boldsymbol{X}, \boldsymbol{y}, \boldsymbol{\varrho}_{k-1}$)
2:    $\boldsymbol{P}_{k-1} \leftarrow \boldsymbol{\Phi}(\boldsymbol{\Theta}_{k-1}, \boldsymbol{X})$                    ▷ Lines 2-4 construct the argument of the gradient in (35)
3:    $\boldsymbol{U}_{k-1} \leftarrow 1/n_{\mathrm{data}} \boldsymbol{P}_{k-1} \boldsymbol{P}_{k-1}^\top$
4:    $\boldsymbol{u}_{k-1} \leftarrow \boldsymbol{U}_{k-1}\boldsymbol{\varrho}_{k-1}$
5:    $\boldsymbol{v}_{k-1} \leftarrow -2/n_{\mathrm{data}} \boldsymbol{P}_{k-1}\boldsymbol{y}$
6:    $\boldsymbol{D} \leftarrow$ BACKWARD $(\boldsymbol{u}_{k-1} + \boldsymbol{v}_{k-1})$                    ▷ Approximate the gradient of (35) using PyTorch library
   BACKWARD (Paszke et al., 2017)
7:    $\boldsymbol{G} \leftarrow \sqrt{2h/\beta} \times \mathrm{randn}_{N \times p}$
8:    $\boldsymbol{\Theta}_k \leftarrow \boldsymbol{\Theta}_{k-1} + h \times \boldsymbol{D} + \boldsymbol{G}$                    ▷ Complete the location update via (35)
9: **end procedure**

---

where $\omega(\cdot) := \int U(\cdot, \tilde{\boldsymbol{\theta}})\varrho(\tilde{\boldsymbol{\theta}})\mathrm{d}\tilde{\boldsymbol{\theta}}$, and $\boldsymbol{\eta}_{k-1}^i := \boldsymbol{\eta}^i(t = (k-1)h)$, $\forall k \in \mathbb{N}$.

To perform this update, EULERMARUYAMA constructs a matrix $\boldsymbol{P}_{k-1}$ whose $(i,j)$th element is $\boldsymbol{P}_{k-1}(i,j) = \Phi(\boldsymbol{x}_j, \boldsymbol{\theta}_{k-1}^i)$. From $\boldsymbol{P}_{k-1}$, we construct $\boldsymbol{v}_{k-1}$ and $\boldsymbol{U}_{k-1}$ as in lines 3 and 5. In line 6 of Algorithm 2, EULERMARUYAMA uses the automatic differentiation module of PyTorch Library, BACKWARD (Paszke et al., 2017), to calculate the gradients needed to update $\boldsymbol{\Theta}_{k-1}$ to $\boldsymbol{\Theta}_k$ $\forall k \in \mathbb{N}$.

Once $\boldsymbol{\Theta}_k$, $\boldsymbol{v}_{k-1}$, and $\boldsymbol{U}_{k-1}$ have been constructed via EULERMARUYAMA, PROXLEARN maps the $N \times 1$ vector $\boldsymbol{\varrho}_{k-1}$ to the proximal update $\boldsymbol{\varrho}_k$.

We next illustrate the implementation of PROXLEARN for binary and multi-class classification case studies. A GitHub repository containing our code for the implementation of these applications can be found at `https://github.com/zalexis12/Proximal-Mean-Field-Learning.git`. Please refer to the Readme file therein for an outline of the structure of our code.

# 4 Case study: binary classification

In this Section, we report numerical results for our first case study, where we apply the proposed PROXLEARN algorithm for binary classification.

For this case study, we perform two implementations on different computing platforms. Our first implementation is on a PC with 3.4 GHz 6-Core Intel Core i5 processor, and 8 GB RAM. For runtime improvement, we then use a Jetson TX2 with a NVIDIA Pascal GPU with 256 CUDA cores, 64-bit NVIDIA Denver and ARM Cortex-A57 CPUs.

## 4.1 WDBC data set

We apply the proposed algorithm to perform a binary classification on the Wisconsin Diagnostic Breast Cancer (henceforth, WDBC) data set available at the UC Irvine machine learning repository (Dua & Graff, 2017). This data set consists of the data of scans from 569 patients. There are $n_x = 30$ features from each scan. Scans are classified as "benign" (which we label as $-1$) or "malignant" (labeled as $+1$).

In (2), we define $\Phi\left(\boldsymbol{x}, \boldsymbol{\theta}_{k-1}^i\right) := a_{k-1}^i \tanh(\langle \boldsymbol{w}_{k-1}^i, \boldsymbol{x}\rangle + b_{k-1}^i) \; \forall i \in [N]$ after $(k-1)$ updates. The parameters $a_{k-1}^i, \boldsymbol{w}_{k-1}^i$ and $b_{k-1}^i$ are the scaling, weight and bias of the $i^{\text{th}}$ sample after $(k-1)$ updates, respectively. Letting $p := n_x + 2$, the parameter vector of the $i^{\text{th}}$ sample after $(k-1)$ updates is

$$\boldsymbol{\theta}_{k-1}^i := \begin{pmatrix} a_{k-1}^i \\ b_{k-1}^i \\ \boldsymbol{w}_{k-1}^i \end{pmatrix} \in \mathbb{R}^p, \quad \forall i \in [N].$$

We set $\varrho_0 \equiv \text{Unif}\left([0.9, 1.1] \times [-0.1, 0.1] \times [-1, 1]^{n_x}\right)$, a uniform joint PDF supported over $n_p = n_x + 2 = 32$ dimensional mean field parameter space.

We use 70% of the entire data set as training data. As discussed in Sec. 3, we learn the mean field parameter distribution via weighted scattered point cloud evolution using PROXLEARN. We then use the confusion matrix method (Visa et al., 2011) to evaluate the accuracy of the obtained model over the test data, which is the remaining 30% of the full data set, containing $n_{\text{test}}$ points.

For each test point $\boldsymbol{x}_{\text{test}} \in \mathbb{R}^{n_x}$, we construct

$$\boldsymbol{\varphi}(\boldsymbol{x}_{\text{test}}) := \begin{pmatrix} \Phi(\boldsymbol{x}_{\text{test}}, \boldsymbol{\theta}_{k-1}^1) \\ \Phi(\boldsymbol{x}_{\text{test}}, \boldsymbol{\theta}_{k-1}^2) \\ \vdots \\ \Phi(\boldsymbol{x}_{\text{test}}, \boldsymbol{\theta}_{k-1}^N) \end{pmatrix} \in \mathbb{R}^N$$

where $\boldsymbol{\theta}_{k-1}^i$ is obtained from the training process. We estimate $f_{\text{MeanField}}$ in (7) in two ways. First, we estimate $f_{\text{MeanField}}$ as a sample average of the elements of $\boldsymbol{\varphi}$. Second, we estimate $f_{\text{MeanField}}$ by numerically approximating the integral in (7) using the propagated samples $\{\boldsymbol{\theta}_{k-1}^i, \varrho_{k-1}^i\}_{i=1}^N$ for $k \in \mathbb{N}$. We refer to these as the "unweighted estimate" and "weighted estimate," respectively. While the first estimate is an empirical average, the second uses the weights $\{\varrho_{k-1}^i\}_{i=1}^N$ obtained from the proposed proximal algorithm. The $f_{\text{MeanField}}$ "unweighted estimate" and "weighted estimate" are then passed through the SOFTMAX and ARGMAX functions respectively, to produce the predicted labels.

## 4.2 Numerical experiments

We set the number of samples $N = 1000$, numerical tolerance $\delta = 10^{-3}$, the maximum number of iterations $L = 300$, and the regularizing parameter $\varepsilon = 1$. Additionally, we set the time step to $h = 10^{-3}$. We run the simulation for different values of the inverse temperature $\beta$, and list the corresponding classification accuracy in Table 1. The "weighted estimate," the ensemble average using proximal updates, produces more accurate results, whereas the "unweighted estimate," the empirical average, is found to be more sensitive to the inverse temperature $\beta$.

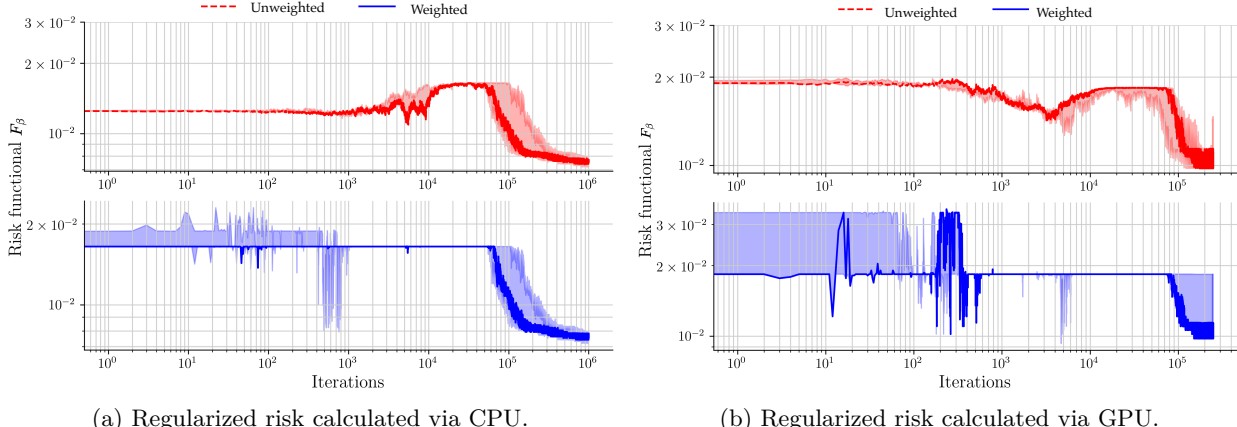

(a) Regularized risk calculated via CPU.     (b) Regularized risk calculated via GPU.

Figure 2: The solid line shows the regularized risk functional $F_\beta$ versus the number of proximal recursions shown for the WDBC dataset with $\beta = 0.05$. The shadow shows the $F_\beta$ variation range for different values of $\beta \in \{0.03, 0.05, 0.07\}$.

Table 1: Classification accuracy of the proposed computational framework for the WDBC Dataset

| $\beta$ | Unweighted | Weighted |
|---|---|---|
| 0.03 | 91.17% | 92.35% |
| 0.05 | 92.94% | 92.94% |
| 0.07 | 78.23% | 92.94% |

For each fixed $\beta$, we perform $10^6$ proximal recursions incurring approx. 33 hours of computational time. Fig. 2a shows the risk functional, computed as the averaged loss over the test data using each of the two estimates described above.

To improve the runtime of our algorithm, we run our code on a Jetson TX2 module, converting data and variables to PyTorch variables.

We begin calculations in Float32, switching to Float64 only when needed to avoid not-a-number (NaN) errors. This switch typically occurs after $2 \times 10^5$ to $3 \times 10^5$ iterations. As shown in Table 2, we train the neural network to a comparable accuracy in only $2.5 \times 10^5$ iterations. The new runtime is around 6% of the original runtime for the CPU-based computation. Fig. 2b shows the risk functional calculated via this updated code. We parallelize these calculations, taking advantage of the GPU capacity of the Jetson TX2.

We utilize these improvements in runtime to additionally experiment with our choice of $\varepsilon$. Table 3 reports the final values of the regularized risk functional $\hat{F}_\beta$ and corresponding runtimes for varying $\varepsilon$. As expected, larger $\varepsilon$ entails more smoothing and lowers runtime. The corresponding final regularized risk values show no significant variations, suggesting numerical stability.

### 4.3 Computational complexity

In this case study, we determine the computational complexity of PROXLEARN as follows. Letting $\boldsymbol{a}_{k-1} := \begin{pmatrix} a_{k-1}^1 & \dots & a_{k-1}^N \end{pmatrix}^\top$, $\boldsymbol{b}_{k-1} := \begin{pmatrix} b_{k-1}^1 & \dots & b_{k-1}^N \end{pmatrix}^\top$, $\boldsymbol{W}_{k-1} := \begin{pmatrix} \boldsymbol{w}_{k-1}^1 & \dots & \boldsymbol{w}_{k-1}^N \end{pmatrix}^\top$, we create the $N \times n_{\text{data}}$ matrix

$$\boldsymbol{P}_{k-1} := \left( \boldsymbol{a}_{k-1} \mathbf{1}^\top \right) \odot \tanh(\boldsymbol{W}_{k-1} \boldsymbol{X}^\top + \boldsymbol{b}_{k-1} \mathbf{1}^\top), \tag{36}$$

which has complexity $O(n_{\text{data}} N n_x)$. The subsequent creation of matrix $\boldsymbol{U}_{k-1}$ in line 3 of Algorithm 2 has complexity $O(n_{\text{data}} N^2)$, and creating $\boldsymbol{v}_{k-1}$ and $\boldsymbol{u}_{k-1}$ takes complexity $O(N n_{\text{data}})$ and $O(N^2)$ respectively.

Table 2: Classification accuracy on Jetson Tx2, after $2.5 \times 10^5$ iterations

| $\beta$ | Unweighted | Weighted | Runtime (hr) |
|---------|------------|----------|--------------|
| 0.03 | 91.18% | 91.18% | 1.415 |
| 0.05 | 91.18% | 92.94% | 1.533 |
| 0.07 | 90.59% | 91.76% | 1.704 |

Table 3: Comparing final $\hat{F}_\beta$ and runtimes for various $\varepsilon$

| $\varepsilon$ | Unweighted Final $\hat{F}_\beta$ | Runtime (s) |
|---------------|----------------------------------|-------------|
| 0.1 | $1.4348 \times 10^{-2}$ | 32863 |
| 0.5 | $1.3740 \times 10^{-2}$ | 11026 |
| 1 | $1.0412 \times 10^{-2}$ | 5022 |
| 5 | $1.1293 \times 10^{-2}$ | 4731 |
| 10 | $9.8849 \times 10^{-3}$ | 4766 |

The complexity in calculating the relevant derivatives of $\boldsymbol{v}_{k-1}$ and $\boldsymbol{u}_{k-1}$ is $O(N^2 n_{\mathrm{data}} n_x)$ (these derivatives are calculated in Appendices B and C). Updating $\boldsymbol{\Theta}_{k-1}$ using these results has complexity of $O(Np) = O(Nn_x)$. Therefore, the process of updating $\boldsymbol{\Theta}_{k-1}$ via EULERMARUYAMA is $O(N^2 n_{\mathrm{data}} n_x)$.

The significant complexity in the remainder of PROXLEARN arises in the construction of matrix $\boldsymbol{C}_k$ in line 3 and the matrix-vector multiplications within the while loop in lines 11, 12.

The creation of the $N \times N$ matrix $\boldsymbol{C}_k$, in which each element is the vector norm of a $n_x \times 1$ vector, is $O(n_x N^2)$. In a worst-case scenario, the while loop runs $L$ times. The operations of leading complexity within the while loop are the multiplications of the $\boldsymbol{\Gamma}_k$ matrix of size $N \times N$ with the $N \times 1$ vectors, which have a complexity of $O(N^2)$. Therefore, the while loop has a complexity of $O(LN^2)$.

Updating $\boldsymbol{\varrho}_{k-1}$ thus has a complexity of $O((n_x + L)N^2)$. In practice, the while loop typically ends far before reaching the maximum number of iterations $L$.

From this analysis, we find that the overall complexity of PROXLEARN is $O(N^2(n_{\mathrm{data}} n_x + L))$. In comparison, the per iteration complexity for JKO-ICNN is $O\left(N_{\mathrm{inner}}(N+1)N_{\mathrm{batch}} + N^3\right)$ where $N_{\mathrm{inner}}$ denotes the number of inner optimization steps, and $N_{\mathrm{batch}}$ denotes the batch size. The per iteration complexity for SWGF is $O\left(N_{\mathrm{inner}} N_{\mathrm{proj}} N_{\mathrm{batch}} \log N_{\mathrm{batch}}\right)$ where $N_{\mathrm{proj}}$ denotes the number of projections to approximate the sliced Wasserstein distance.

## 4.4 Comparisons to existing results

From Fig. 2, we observe that there is a significant burn in period for the risk functional curves. These trends in learning curves agree with those reported in (Mei et al., 2018). In particular, (Mei et al., 2018, Fig. 3) and Fig. 7.3 in that reference's *Supporting Information*, show convergence trends very similar to our Fig. 2: slow decay until approx. $10^5$ iterations and then a significant speed up. The unusual convergence trend was explicitly noted in (Mei et al., 2018): "We observe that SGD converges to a network with very small risk, but this convergence has a nontrivial structure and presents long flat regions". It is interesting to note that (Mei et al., 2018) considered an experiment that allowed rotational symmetry and simulated the radial (i.e., with one spatial dimension) discretized PDE, while we used the proposed proximal recursion directly in the neuron population ensemble to solve the PDE IVP, i.e., similar convergence trends were observed using different numerical methods applied to the same mean field PDE IVP. This makes us speculate that the convergence trend is specific to the mean field PDE dynamics itself, and is less about the particular numerical algorithm. This observation is consistent with recent works such as (Wojtowytsch & Weinan, 2020) which investigate the mean field learning dynamics in two layer ReLU networks and in that setting, show that the learning can indeed be slow depending on the target function.

Table 4: Comparison of average classification accuracy from Bonet et al. (2022, Table 1) to our algorithm, ProxLearn

| Dataset | JKO-ICNN | SWGF + RealNVP | ProxLearn, Weighted | ProxLearn, Unweighted |
|---------|----------|----------------|---------------------|------------------------|
| Banana | $0.550 \pm 10^{-2}$ | $0.559 \pm 10^{-2}$ | $0.551 \pm 10^{-2}$ | $0.535 \pm 5 \cdot 10^{-2}$ |
| Diabetes | $0.777 \pm 7 \cdot 10^{-3}$ | $0.778 \pm 2 \cdot 10^{-3}$ | $0.736 \pm 2 \cdot 10^{-2}$ | $0.731 \pm 10^{-2}$ |
| Twonorm | $0.981 \pm 2 \cdot 10^{-4}$ | $0.981 \pm 6 \cdot 10^{-4}$ | $0.972 \pm 2 \cdot 10^{-3}$ | $0.972 \pm 2 \cdot 10^{-3}$ |

As a first study, our numerical results achieve reasonable classification accuracy compared to the state-of-art, even though our proposed meshless proximal algorithm is very different from the existing implementations. We next compare the numerical performance with existing methods as in Mokrov et al. (2021) and Bonet et al. (2022). We apply our proximal algorithm for binary classification to three datasets also considered in Mokrov et al. (2021) and Bonet et al. (2022): the banana, diabetes, and twonorm datasets.

The banana dataset consists of 5300 data points, each with $n_x = 2$ features, which we rescale to lie between 0 and 8. We set $\beta = 0.05$, draw our initial weights $\boldsymbol{w}$ from $\mathrm{Unif}\left([-2,2]^{n_x}\right)$ and bias $b$ from $\mathrm{Unif}\left([-0.3,0.3]\right)$, and set $\boldsymbol{\varrho}_0 \equiv \mathrm{Unif}\left(0,1000\right)$. We run our code for 3500 iterations, splitting the data evenly between test and training data.

The diabetes dataset consists of $n_x = 8$ features from each of 768 patients. Based on our experimental results, we make the following adjustments to our algorithm: we redefine $\beta = 0.65$, $\boldsymbol{\varrho}_0 \equiv \mathrm{Unif}\left(0,1000\right)$, and draw our initial weights $\boldsymbol{w}$ from $\mathrm{Unif}\left([-2,2]^{n_x}\right)$. We rescale the data to lie between 0 and 1, and use half of the dataset for training purposes, and the remainder as test data. In this case, we run our code for $4.99 \times 10^5$ iterations.

The twonorm dataset consists of 7,400 samples drawn from two different normal distributions, with $n_x = 20$ features. We again consider 50% of the same as training data and used the remaining 50% as test data, and rescale the given data by a factor of 8. Based on our empirical observations, we redefine $\beta = 1.95$, and once more draw our initial weights $\boldsymbol{w}$ from $\mathrm{Unif}\left([-2,2]^{n_x}\right)$ and set $\boldsymbol{\varrho}_0 \equiv \mathrm{Unif}\left(0,1000\right)$. In this case, we perform $10^4$ proximal recursions in each separate run.

We run our code five times for each of the three datasets under consideration and compute "unweighted estimates" and "weighted estimates" in each case, as described above. These estimates assign each data point a value: negative values predict the label as 0, while positive values predict the label as 1. From these results, we calculate the weighted and unweighted accuracy by finding the percentage of predicted test labels that match the actual test labels. The average accuracy over all five runs is reported in Table 4, alongside the results reported in Bonet et al. (2022, Table 1). We achieve comparable accuracy to these recent results.

# 5 Case study: multi-class classification

We next apply the proposed proximal algorithm to a ten-class classification problem using the Semeion Handwritten Digit (hereafter SHD) Data Set (Dua & Graff, 2017). This numerical experiment is performed on the aforementioned Jetson TX2.

## 5.1 SHD data set

The SHD Data Set consists of 1593 handwritten digits. By viewing each digit as $16 \times 16$ pixel image, each image is represented by $n_x = 16^2 = 256$ features. Each feature is a boolean value indicating whether a particular pixel is filled. We subsequently re-scale these features such that $\boldsymbol{x}_i \in \{-1,1\}^{n_x}$.

## 5.2 Adaptations to ProxLearn for multi-class case

To apply PROXLEARN for a multi-class case, we make several adaptations. For instance, rather than attempting to determine $f(\boldsymbol{x}) \approx y$, we redefine $f(\boldsymbol{x})$ to represent a mapping of input data to the predicted likelihood of the correct label. We therefore redefine the variables, parameters, and risk function as follows.

Each label is represented by a $1 \times 10$ vector of booleans, stored in a $n_{\text{data}} \times 10$ matrix $\boldsymbol{Y}$ where $Y_{i,j} = 1$ if the $i^{\text{th}}$ data point $\boldsymbol{x}_i$ has label $j$, and $Y_{i,j} = 0$ otherwise.

We construct the $N \times n_{\text{data}}$ matrix $\boldsymbol{P}_{k-1}$ by defining the $(j, i)$ element of $\boldsymbol{P}_{k-1}$ as

$$\boldsymbol{P}_{k-1}(j, i) := \boldsymbol{\Phi}(\boldsymbol{\theta}_{k-1}^j, \boldsymbol{X}(i,:), \boldsymbol{Y}(i,:))$$

$$:= \left\langle \text{SOFTMAX}(\boldsymbol{X}(i,:)(\boldsymbol{\theta}_{k-1}^j)^\top), (\boldsymbol{Y}(i,:))^\top \right\rangle \tag{37}$$

where $\langle \cdot, \cdot \rangle$ denotes the standard Euclidean inner product. The SOFTMAX function in (37) produces a $10 \times 1$ vector of non-negative entries that sum to 1. By taking the inner product of this vector with the Boolean vector $\boldsymbol{Y}(i,:)$, we define $\boldsymbol{P}_{k-1}(j, i) = \boldsymbol{\Phi}(\boldsymbol{\theta}_{k-1}^j, \boldsymbol{X}(i,:), \boldsymbol{Y}(i,:))$ as the perceived likelihood that the data point $i$ is labeled correctly by sample $j$. As our model improves, this value approaches 1, which causes the probability of an incorrect label to drop.

As this newly defined $\boldsymbol{P}_{k-1}$ does not call for bias or scaling, the weights alone are stored in the $N \times p$ matrix $\boldsymbol{\Theta}_{k-1}$. In this case, $p := 10n_x$, as each of the $n_x$ features requires a distinct weight for each of the ten labels. For convenience, we reshape $\boldsymbol{\Theta}_{k-1} = (\boldsymbol{\theta}_{k-1}^1, ..., \boldsymbol{\theta}_{k-1}^N)$, where each $\boldsymbol{\theta}_{k-1}^i$ is a $10 \times n_x$ matrix. Therefore, $\boldsymbol{\Theta}_{k-1}$ is a $10 \times n_x \times N$ tensor.

We redefine the unregularized risk to reflect our new $\boldsymbol{\Phi}$ as follows:

$$F(\rho) := \mathbb{E}_{(\boldsymbol{x}, \boldsymbol{y})} \left( 1 - \int_{\mathbb{R}^p} \Phi(\boldsymbol{x}, \boldsymbol{y}, \boldsymbol{\theta}) \rho(\boldsymbol{\theta}) \mathrm{d}\boldsymbol{\theta} \right)^2 . \tag{38}$$

Expanding the above, we arrive at a form that resembles (8), now with the following adjusted definitions:

$$F_0 := 1, \tag{39}$$

$$V(\boldsymbol{\theta}) := \mathbb{E}_{(\boldsymbol{x}, \boldsymbol{y})}[-2\Phi(\boldsymbol{\theta}, \boldsymbol{x}, \boldsymbol{y})], \tag{40}$$

$$U(\boldsymbol{\theta}, \tilde{\boldsymbol{\theta}}) := \mathbb{E}_{(\boldsymbol{x}, \boldsymbol{y})}[\Phi(\boldsymbol{\theta}, \boldsymbol{x}, \boldsymbol{y})\Phi(\tilde{\boldsymbol{\theta}}, \boldsymbol{x}, \boldsymbol{y})]. \tag{41}$$

We use the regularized risk functional $F_\beta$ as in (12) where $F$ now is given by (38). Due to the described changes in the structure of $\boldsymbol{\Theta}_{k-1}$, the creation of $\boldsymbol{C}_k$ in line 3 of PROXLEARN results in a $10 \times N \times N$ tensor, which we then sum along the ten element axis, returning an $N \times N$ matrix.

Finally, we add a scaling in line 7 of EULERMARUYAMA, scaling the noise by a factor of $1/100$.

## 5.3 Numerical experiments

With the adaptations mentioned above, we set the inverse temperature $\beta = 0.5$, $\epsilon = 10$, the step size $h = 10^{-3}$, and $N = 100$. We draw the initial weights from $\text{Unif}\left([-1, 1]^{10n_x}\right)$. We take the first $n_{\text{data}} = 1000$ images as training data, reserving the remaining $n_{\text{test}} = 593$ images as test data, and execute 30 independent runs of our code, each for $10^6$ proximal recursions.

To evaluate the training process, we create a matrix $\boldsymbol{P}_{k-1}^{\text{test}} \in \mathbb{R}^{N \times n_{\text{test}}}$, using test data rather than training data, but otherwise defined as in (37). We then calculate a weighted approximation of $F_\beta$:

$$F_\beta \approx \frac{1}{n_{\text{test}}} \left\| \mathbf{1} - (\boldsymbol{P}_{k-1}^{\text{test}})^\top \boldsymbol{\varrho} \right\|_2^2, \tag{42}$$

and an unweighted approximation:

$$F_\beta \approx \frac{1}{n_{\text{test}}} \left\| \mathbf{1} - \frac{1}{N}(\boldsymbol{P}_{k-1}^{\text{test}})^\top \mathbf{1} \right\|_2^2, \tag{43}$$

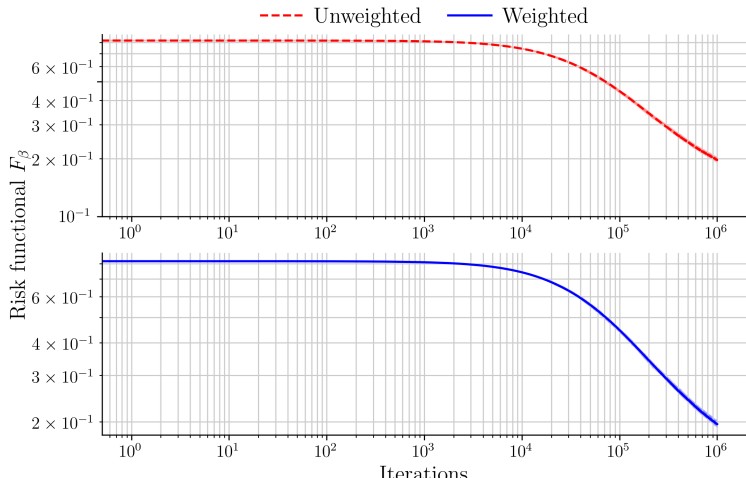

Figure 3: The solid line shows the average regularized risk functional $F_\beta$ versus the number of proximal recursions shown for the Semeion dataset with $\beta = 0.5$. The narrow shadow shows the $F_\beta$ variation range with the same $\beta$ using the results of 30 independent runs, each starting from the same initial point cloud $\{\boldsymbol{\theta}_0^i, \varrho_0^i\}_{i=1}^N$.

which we use to produce the risk and weighted risk log-log plots shown in Fig. 3.

Notably, despite the new activation function and the adaptations described above, our algorithm produces similar risk plots in the binary and multi-class cases. The run time for $10^6$ iterations is approximately 5.3 hours.

To evaluate our multi-class model, we calculate the percentage of accurately labeled test data by first taking ARGMAX $(\boldsymbol{X}_{\text{test}}\boldsymbol{\Theta}_{k-1})$ along the ten dimensional axis, to determine the predicted labels for each test data point using each sample of $\boldsymbol{\Theta}_{k-1}$. We then compare these predicted labels with the actual labels. We achieved 61.079% accuracy for the test data, and 75.330% accuracy for the training data.

### 5.4 Updated computational complexity of ProxLearn

In the binary case, the creation of matrix $\boldsymbol{C}_k$ requires $O(n_x N^2)$ flops. In the case of multi-class classification concerning $m$ classes, $\boldsymbol{C}_k$ is redefined as the sum of $m$ such matrices. Therefore, creating the updated matrix $\boldsymbol{C}_k$ takes $O(mn_x N^2)$ flops. Thus, updating $\boldsymbol{\varrho}_{k-1}$ is of complexity $O((mn_x + L)N^2)$. The complexity of EULERMARUYAMA can be generalized from the discussion in Sec. 4.

## 6 Conclusions

### 6.1 Summary

This work presents a proximal mean field learning algorithm to train a shallow neural network in the over-parameterized regime. The proposed algorithm is meshless, non-parametric and implements the Wasserstein proximal recursions realizing the gradient descent of entropic-regularized risk. Numerical case studies in binary and multi-class classification demonstrate that the ideas of mean field learning can be attractive as computational framework, beyond purely theoretical interests. Our contribution should be of interest to other learning and variational inference tasks such as the policy optimization and adversarial learning.

We clarify that the proposed algorithm is specifically designed for a neural network with single hidden layer in large width regime. For multi-hidden layer neural networks, the mean field limit in the sense of width as pursued here, is relatively less explored for the training dynamics. In the multiple hidden layer setting, theoretical understanding of the limits is a frontier of current research; see e.g., Fang et al. (2021); Sirignano & Spiliopoulos (2022). Extending these ideas to design variants of proximal algorithms requires new lines of thought, and as such, is out-of-scope of this paper. In the following, we outline the scope for such future work.

### 6.2 Scope for future work

Existing efforts to generalize the theoretical results for the mean field limit as in this work, from single to multi-hidden layer networks, have been pursued in two different limiting sense. One line of investigations (Sirignano & Spiliopoulos, 2020; 2022; Yu & Spiliopoulos, 2023) take the infinite width limit one hidden layer at a time while holding the (variable) widths of other hidden layers fixed. More precisely, if the $i$th hidden layer has $N_i$ neurons, then the limit is taken by first normalizing that layer's output by $N_i^{\gamma_i}$ for some fixed $\gamma_i \in [1/2, 1]$ and then letting $N_i \to \infty$ for an index $i$ while holding other $N_j$'s fixed and finite, $j \neq i$.

A different line of investigations (Araújo et al., 2019; Nguyen, 2019) consider the limit where the widths of all hidden layers simultaneously go to infinity. In this setting, the population distribution over the joint (across hidden layers) parameter space is shown to evolve under SGD as per a McKean-Vlasov type nonlinear IVP; see (Araújo et al., 2019, Def. 4.4 and Sec. 5). We anticipate that the proximal recursions proposed herein can be extended to this setting by effectively lifting the Wasserstein gradient flow to the space of measure-valued paths. Though not quite the same, but this is similar in spirit to how classical bi-mariginal Schrödinger bridge problems (Léonard, 2012; Chen et al., 2021) have been generalized to multi-marginal settings over the path space and have led to significant algorithmic advances in recent years (Haasler et al., 2021; Carlier, 2022; Chen et al., 2023). Pursuing such ideas for designing proximal algorithms in the multiple hidden layer case will comprise our future work.

## Acknowledgment

This work was supported by NSF award 2112755. The authors acknowledge the reviewers' perceptive feedback to help improve this paper.

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

## A  Proof of Theorem 1

We provide the formal statement followed by the proof.

**Theorem 1.** *Consider the regularized risk functional (12) wherein $F$ is given by (8)-(9). Let $\rho(t, \boldsymbol{\theta})$ solve the IVP (14), and let $\{\varrho_{k-1}\}_{k \in \mathbb{N}}$ be the sequence generated by (17) with $\varrho_0 \equiv \rho_0$. Define the interpolation $\varrho_h : [0, \infty) \times \mathbb{R}^p \mapsto [0, \infty)$ as*

$$\varrho_h(t, \boldsymbol{\theta}) := \varrho_{k-1}(h, \boldsymbol{\theta}) \,\forall t \in [(k-1)h, kh), \ k \in \mathbb{N}.$$

*Then $\varrho_h(t, \boldsymbol{\theta}) \xrightarrow{h \downarrow 0} \rho(t, \boldsymbol{\theta})$ in $L^1(\mathbb{R}^p)$.*

*Proof.* Our proof follows the general development in Laborde (2017, Sec. 12.3). In the following, we sketch the main ideas.

We have the semi-implicit free energy

$$\hat{F}_\beta(\varrho, \varrho_{k-1}) = \underbrace{\mathbb{E}_\varrho\left[F_0 + V(\boldsymbol{\theta}) + \int_{\mathbb{R}^p} U(\boldsymbol{\theta}, \tilde{\boldsymbol{\theta}})\varrho_{k-1}(\tilde{\boldsymbol{\theta}})\mathrm{d}\tilde{\boldsymbol{\theta}}\right]}_{=:\mathcal{V}_{\mathrm{advec}}(\varrho)} + \beta^{-1}\mathbb{E}_\varrho\left[\log\varrho\right], \quad k \in \mathbb{N},$$

wherein the summand

$$\mathcal{V}_{\mathrm{advec}}(\varrho) := \mathbb{E}_\varrho\left[F_0 + V(\boldsymbol{\theta}) + \int_{\mathbb{R}^p} U(\boldsymbol{\theta}, \tilde{\boldsymbol{\theta}})\varrho_{k-1}(\tilde{\boldsymbol{\theta}})\mathrm{d}\tilde{\boldsymbol{\theta}}\right]$$

is linear in $\varrho$, and contributes as an effective advection potential energy. The remaining summand $\beta^{-1}\mathbb{E}_\varrho\left[\log\varrho\right]$ results in from diffusion regularization and contributes as an internal energy term.

We note from equation 9 that the functional $\mathcal{V}_{\mathrm{advec}}(\varrho)$ is lower bounded for all $\varrho \in \mathcal{P}_2\left(\mathbb{R}^p\right)$. Furthermore, $\mathcal{V}_{\mathrm{advec}}(\varrho)$ and $\nabla\mathcal{V}_{\mathrm{advec}}(\varrho)$ are uniformly Lipschitz continuous, i.e., there exists $C_1 > 0$ such that

$$\|\nabla\mathcal{V}_{\mathrm{advec}}(\varrho)\|_{L^\infty(\mathbb{R}^p)} + \|\nabla^2\mathcal{V}_{\mathrm{advec}}(\varrho)\|_{L^\infty(\mathbb{R}^p)} \le C_1$$

for all $\varrho \in \mathcal{P}_2\left(\mathbb{R}^p\right)$ where the constant $C_1 > 0$ is independent of $\varrho$, and $\nabla^2$ denotes the Euclidean Hessian operator.

Moreover, there exists $C_2 > 0$ such that for all $\varrho, \tilde{\varrho} \in \mathcal{P}_2(\mathbb{R}^p)$, we have

$$\|\nabla\mathcal{V}_{\mathrm{advec}}(\varrho) - \nabla\mathcal{V}_{\mathrm{advec}}(\tilde{\varrho})\|_{L^\infty(\mathbb{R}^p)} \le C_2 W_2\left(\varrho, \tilde{\varrho}\right).$$

Thus, $\mathcal{V}_{\mathrm{advec}}(\varrho)$ satisfy the hypotheses in Laborde (2017, Sec. 12.2).

For $t \in [0, T]$, we say that $\rho(t, \boldsymbol{\theta}) \in C\left([0, T], \mathcal{P}_2\left(\mathbb{R}^p\right)\right)$ is a weak solution of the IVP equation 14 if for any smooth compactly supported test function $\varphi \in C_c^\infty\left([0, \infty) \times \mathbb{R}^p\right)$, we have

$$\int_0^\infty \int_{\mathbb{R}^p}\left(\left(\frac{\partial\varphi}{\partial t} - \langle\nabla\varphi, \nabla\mathcal{V}_{\mathrm{advec}}(\varrho)\rangle\right)\rho + \beta^{-1}\rho\Delta\varphi\right)\mathrm{d}\boldsymbol{\theta}\mathrm{d}t = -\int_{\mathbb{R}^p}\varphi(t = 0, \boldsymbol{\theta})\rho_0(\boldsymbol{\theta}). \tag{44}$$

Following Laborde (2017, Sec. 12.2), under the stated hypotheses on $\mathcal{V}_{\mathrm{advec}}(\varrho)$, there exists weak solution of the IVP equation 14 that is continuous w.r.t. the $W_2$ metric.

The remaining of the proof follows the outline below.

- Using the Dunford-Pettis' theorem, establish that the sequence of functions $\{\varrho_k(h, \boldsymbol{\theta})\}_{k\in\mathbb{N}}$ solving equation 17 is unique.

- Define the interpolation $\varrho_h(t) := \varrho_k(h, \boldsymbol{\theta})$ if $t \in ((k-1)h, kh]$ for all $k \in \mathbb{N}$. Then establish that $\varrho_h(t)$ solves a discrete approximation of equation 44.

- Finally combine the gradient estimates and pass to the limit $h \downarrow 0$, to conclude that $\rho_h(t)$ in this limit solves converges to the weak solution of equation 44 in strong $L^1(\mathbb{R}^p)$ sense.

For the detailed calculations on the passage to the limit, we refer the readers to Laborde (2017, Sec. 12.5). $\quad\square$

## B   Expressions involving the derivatives of $v$

We define matrices

$$\boldsymbol{T} := \tanh\left(\boldsymbol{W}\boldsymbol{X}^\top + \boldsymbol{b}\boldsymbol{1}^\top\right),$$
$$\boldsymbol{S} := \mathrm{sech}^2\left(\boldsymbol{W}\boldsymbol{X}^\top + \boldsymbol{b}\boldsymbol{1}^\top\right),$$

where $\mathbf{1}$ is a vector of all ones of size $n_{\text{data}} \times 1$, and the functions $\tanh(\cdot)$ and $\text{sech}^2(\cdot)$ are elementwise. Notice that $\boldsymbol{T}, \boldsymbol{S} \in \mathbb{R}^{N \times n_{\text{data}}}$. Then

$$\boldsymbol{v} = -\frac{2}{n_{\text{data}}} \boldsymbol{a} \odot \left( \tanh(\boldsymbol{W} \boldsymbol{X}^\top + \boldsymbol{b} \mathbf{1}^\top) \boldsymbol{y} \right) = -\frac{2}{n_{\text{data}}} \boldsymbol{a} \odot (\boldsymbol{T} \boldsymbol{y}).$$

**Proposition 2.** *With the above notations in place, we have*

$$\frac{\partial \boldsymbol{v}}{\partial \boldsymbol{a}} \mathbf{1} = \sum_{k=1}^{N} \frac{\partial \boldsymbol{v}_k}{\partial \boldsymbol{a}} = -\frac{2}{n_{\text{data}}} \boldsymbol{T} \boldsymbol{y}, \tag{45}$$

*and*

$$\frac{\partial \boldsymbol{v}}{\partial \boldsymbol{b}} \mathbf{1} = \sum_{k=1}^{N} \frac{\partial \boldsymbol{v}_k}{\partial \boldsymbol{b}} = -\frac{2}{n_{\text{data}}} \boldsymbol{a} \odot \boldsymbol{S} \boldsymbol{y}. \tag{46}$$

*Furthermore,*

$$\sum_{k=1}^{N} \frac{\partial \boldsymbol{v}_k}{\partial \boldsymbol{W}} = -\frac{2}{n_{\text{data}}} \left[ (\boldsymbol{a} \mathbf{1}^\top) \odot \left( \boldsymbol{S} \left( \boldsymbol{X} \odot \boldsymbol{y} \mathbf{1}^\top \right) \right) \right]. \tag{47}$$

*Proof.* The $k^{\text{th}}$ element of $\boldsymbol{v}$ is $\boldsymbol{v}_k = -\frac{2}{n_{\text{data}}} \boldsymbol{a}_k \sum_{i=1}^{n_{\text{data}}} [\boldsymbol{T}_{k,i} \boldsymbol{y}_i]$. Thus,

$$\frac{\partial \boldsymbol{v}_k}{\partial \boldsymbol{a}_j} = \begin{cases} 0 & \text{for} \quad k \neq j, \\ -\frac{2}{n_{\text{data}}} \sum_{i=1}^{n_{\text{data}}} [\boldsymbol{T}_{k,i} \boldsymbol{y}_i] & \text{for} \quad k = j. \end{cases}$$

So the matrix $\frac{\partial \boldsymbol{v}}{\partial \boldsymbol{a}}$ is diagonal, and

$$\left[ \frac{\partial \boldsymbol{v}}{\partial \boldsymbol{a}} \mathbf{1} \right]_k = -\frac{2}{n_{\text{data}}} \sum_{i=1}^{n_{\text{data}}} [\boldsymbol{T}_{k,i} \boldsymbol{y}_i] = -\frac{2}{n_{\text{data}}} [\boldsymbol{T} \boldsymbol{y}]_k.$$

Hence, we obtain

$$\frac{\partial \boldsymbol{v}}{\partial \boldsymbol{a}} \mathbf{1} = -\frac{2}{n_{\text{data}}} \boldsymbol{T} \boldsymbol{y},$$

which is (45).

On the other hand,

$$\frac{\partial \boldsymbol{v}_k}{\partial \boldsymbol{b}_k} = \frac{\partial}{\partial \boldsymbol{b}_k} \left[ -\frac{2}{n_{\text{data}}} \boldsymbol{a}_k \sum_{i=1}^{n_{\text{data}}} [\boldsymbol{T}_{k,i} \boldsymbol{y}_i] \right]$$

$$= -\frac{2}{n_{\text{data}}} \boldsymbol{a}_k \sum_{i=1}^{n_{\text{data}}} \frac{\partial}{\partial \boldsymbol{b}_k} [\boldsymbol{T}_{k,i} \boldsymbol{y}_i].$$

Note that

$$\frac{\partial}{\partial \boldsymbol{b}_k} [\boldsymbol{T}_{k,i} \boldsymbol{y}_i] = \frac{\partial}{\partial \boldsymbol{b}_k} \tanh \left( \sum_{j=1}^{n_x} (\boldsymbol{W}_{k,j} \boldsymbol{X}_{i,j}) + \boldsymbol{b}_k \right) \boldsymbol{y}_i$$

$$= \text{sech}^2 \left( \sum_{j=1}^{n_x} (\boldsymbol{W}_{k,j} \boldsymbol{X}_{i,j}) + \boldsymbol{b}_k \right) \boldsymbol{y}_i = \boldsymbol{S}_{k,i} \boldsymbol{y}_i.$$

Therefore,

$$\frac{\partial \boldsymbol{v}_k}{\partial \boldsymbol{b}_j} = \begin{cases} 0 & \text{for} \quad k \neq j, \\ -\frac{2}{n_{\text{data}}} \boldsymbol{a}_k \sum_{i=1}^{n_{\text{data}}} [\boldsymbol{S}_{k,i} \boldsymbol{y}_i] & \text{for} \quad k = j. \end{cases}$$

As the matrix $\frac{\partial \boldsymbol{v}}{\partial \boldsymbol{b}}$ is diagonal, we get

$$\left[ \frac{\partial \boldsymbol{v}}{\partial \boldsymbol{b}} \mathbf{1} \right]_k = -\frac{2}{n_{\text{data}}} \boldsymbol{a}_k \sum_{i=1}^{n_{\text{data}}} [\boldsymbol{S}_{k,i} \boldsymbol{y}_i] = -\frac{2}{n_{\text{data}}} \boldsymbol{a}_k \left[ \boldsymbol{S}\boldsymbol{y} \right]_k,$$

and so

$$\frac{\partial \boldsymbol{v}}{\partial \boldsymbol{b}} \mathbf{1} = -\frac{2}{n_{\text{data}}} \boldsymbol{a} \odot \boldsymbol{S}\boldsymbol{y},$$

which is indeed (46).

Likewise, we take an element-wise approach to the derivatives with respect to weights $\boldsymbol{W}_{k,m}$. Note that such a weight $\boldsymbol{W}_{k,m}$ will only appear in the $k$th element of $\boldsymbol{v}$, and so we only need to compute

$$\frac{\partial \boldsymbol{v}_k}{\partial \boldsymbol{W}_{k,m}} = -\frac{2}{n_{\text{data}}} \boldsymbol{a}_k \sum_{i=1}^{n_{\text{data}}} \frac{\partial}{\partial \boldsymbol{W}_{k,m}} \left[ \boldsymbol{T}_{k,i} \boldsymbol{y}_i \right].$$

Since

$$\frac{\partial}{\partial \boldsymbol{W}_{k,m}} \left[ \boldsymbol{T}_{k,i} \boldsymbol{y}_i \right] = \frac{\partial}{\partial \boldsymbol{W}_{k,m}} \tanh \left( \sum_{j=1}^{n_x} (\boldsymbol{W}_{k,j} \boldsymbol{X}_{j,i}) + \boldsymbol{b}_k \right) \boldsymbol{y}_i$$

$$= \text{sech}^2 \left( \sum_{j=1}^{n_x} (\boldsymbol{W}_{k,j} \boldsymbol{X}_{i,j}) + \boldsymbol{b}_k \right) \boldsymbol{X}_{i,m} \boldsymbol{y}_i = \boldsymbol{S}_{k,i} \boldsymbol{X}_{i,m} \boldsymbol{y}_i,$$

we have

$$\frac{\partial \boldsymbol{v}_k}{\partial \boldsymbol{W}_{m,j}} = \begin{cases} 0 & \text{for} \quad k \neq m, \\ -\frac{2}{n_{\text{data}}} \boldsymbol{a}_k \sum_{i=1}^{n_{\text{data}}} [\boldsymbol{S}_{k,i} \boldsymbol{X}_{i,m} \boldsymbol{y}_i] & \text{for} \quad k = m. \end{cases}$$

Thus,

$$\sum_{k=1}^{N} \frac{\partial \boldsymbol{v}_k}{\partial \boldsymbol{W}_{m,j}} = -\frac{2}{n_{\text{data}}} \boldsymbol{a}_m \sum_{i=1}^{n_{\text{data}}} [\boldsymbol{S}_{m,i} \boldsymbol{X}_{i,m} \boldsymbol{y}_i]$$

$$= -\frac{2}{n_{\text{data}}} \boldsymbol{a}_m \left[ \boldsymbol{S} \left( \boldsymbol{X} \odot (\boldsymbol{y}\mathbf{1}^\top) \right) \right]_{m,j}.$$

Therefore, considering $\mathbf{1} \in \mathbb{R}^{n_x}$, we write

$$\sum_{k=1}^{N} \frac{\partial \boldsymbol{v}_k}{\partial \boldsymbol{W}} = -\frac{2}{n_{\text{data}}} \left[ (\boldsymbol{a}\mathbf{1}^\top) \odot \left( \boldsymbol{S} \left( \boldsymbol{X} \odot \boldsymbol{y}\mathbf{1}^\top \right) \right) \right],$$

thus arriving at (47). This completes the proof. $\square$

## C  Expressions involving the derivatives of $\boldsymbol{u}$

Expressions involving the derivatives of $\boldsymbol{u}$, are summarized in the Proposition next. These results find use in Sec. 4. We start by noting that

$$\boldsymbol{u} = \frac{1}{n_{\text{data}}} (\mathbf{1}^\top \boldsymbol{a} \odot \boldsymbol{T})(\mathbf{1}^\top \boldsymbol{a} \odot \boldsymbol{T})^\top \boldsymbol{\rho}$$

$$= \frac{1}{n_{\text{data}}} (\mathbf{1}^\top \boldsymbol{a} \odot \boldsymbol{T})(\boldsymbol{a}^\top \mathbf{1} \odot \boldsymbol{T}^\top) \boldsymbol{\rho}.$$

**Proposition 3.** *With the above notations in place, we have*

$$\underbrace{\frac{\partial \boldsymbol{u}}{\partial \boldsymbol{a}}}_{N \times N} \underbrace{\boldsymbol{1}}_{N \times 1} = \frac{1}{n_{\text{data}}} \left[ \left( (\boldsymbol{\varrho}\boldsymbol{a}^\top) \odot (\boldsymbol{T}\boldsymbol{T}^\top) \right) \boldsymbol{1} + \boldsymbol{1} \left( \boldsymbol{a} \odot \boldsymbol{\varrho} \right)^\top \boldsymbol{T}\boldsymbol{T}^\top \boldsymbol{1} \right], \tag{48}$$

*and*

$$\underbrace{\frac{\partial \boldsymbol{u}}{\partial \boldsymbol{b}}}_{N \times N} \underbrace{\boldsymbol{1}}_{N \times 1} = \frac{1}{n_{\text{data}}} \left[ \left( (\boldsymbol{a}\boldsymbol{1}^\top) \odot (\boldsymbol{S}\boldsymbol{T}^\top) \odot \left( \boldsymbol{1} (\boldsymbol{a} \odot \boldsymbol{\varrho})^\top \right) \right) \boldsymbol{1} \right.$$

$$\left. + \left( (\boldsymbol{1}\boldsymbol{a}^\top) \odot (\boldsymbol{S}\boldsymbol{T}^\top) \odot ((\boldsymbol{a} \odot \boldsymbol{\varrho})\boldsymbol{1}^\top) \right) \boldsymbol{1} \right]. \tag{49}$$

*Furthermore,*

$$\sum_{k=1}^{N} \frac{\partial \boldsymbol{u}}{\partial \boldsymbol{W}_{i,j}} = \frac{1}{n_{\text{data}}} \sum_{k=1}^{N} \sum_{m=1}^{n_{\text{data}}} a_i a_k (\varrho_i + \varrho_k) T_{k,m} S_{i,m} X_{m,j}. \tag{50}$$

*Proof.* Letting $\boldsymbol{t}_i^\top$ denote the $i$th row of $\boldsymbol{T}$, we rewrite $\boldsymbol{u}$ as follows:

$$\boldsymbol{u} = \frac{1}{n_{\text{data}}} \begin{bmatrix} a_1 \boldsymbol{t}_1^\top \\ \vdots \\ a_N \boldsymbol{t}_N^\top \end{bmatrix} \left[ \rho_1 a_1 \boldsymbol{t}_1 + \ldots + \rho_N a_N \boldsymbol{t}_N \right]$$

$$= \frac{1}{n_{\text{data}}} \begin{bmatrix} a_1 \boldsymbol{t}_1^\top (\rho_1 a_1 \boldsymbol{t}_1 + \ldots + \rho_N a_N \boldsymbol{t}_N) \\ \vdots \\ a_N \boldsymbol{t}_N^\top (\rho_1 a_1 \boldsymbol{t}_1 + \ldots + \rho_N a_N \boldsymbol{t}_N) \end{bmatrix}.$$

For $i \neq k$, we thus have

$$\frac{\partial \boldsymbol{u}_i}{\partial \boldsymbol{a}_k} = \frac{1}{n_{\text{data}}} a_i \boldsymbol{t}_i^\top (\rho_k \boldsymbol{t}_k) = \frac{1}{n_{\text{data}}} a_i \rho_k \boldsymbol{t}_i^\top \boldsymbol{t}_k.$$

Likewise, for $i = k$, we have

$$\frac{\partial \boldsymbol{u}_k}{\partial \boldsymbol{a}_k} = \frac{1}{n_{\text{data}}} a_k \rho_k \boldsymbol{t}_k^\top \boldsymbol{t}_k + \frac{1}{n_{\text{data}}} \boldsymbol{t}_k^\top (\rho_1 a_1 \boldsymbol{t}_1 + \ldots + \rho_N a_N \boldsymbol{t}_N).$$

Combining the above, we obtain $\frac{\partial \boldsymbol{u}}{\partial \boldsymbol{a}}$, and hence (48) follows.

On the other hand, for $i \neq k$, we have

$$\frac{\partial \boldsymbol{u}_i}{\partial \boldsymbol{b}_k} = \frac{1}{n_{\text{data}}} a_i \boldsymbol{t}_i^\top \rho_k a_k \frac{\partial \boldsymbol{t}_k}{\partial \boldsymbol{b}_k} = \frac{1}{n_{\text{data}}} a_i \boldsymbol{t}_i^\top \rho_k a_k \boldsymbol{s}_k,$$

and for $i = k$, we obtain

$$\frac{\partial \boldsymbol{u}_i}{\partial \boldsymbol{b}_k} = \frac{1}{n_{\text{data}}} a_i \left( \frac{\partial \boldsymbol{t}_k}{\partial \boldsymbol{b}_k} \right)^\top (\rho_1 a_1 \boldsymbol{t}_1 + \ldots + \rho_N a_N \boldsymbol{t}_N)$$

$$+ \frac{1}{n_{\text{data}}} a_k \boldsymbol{t}_k^\top \rho_k a_k \frac{\partial \boldsymbol{t}_k}{\partial \boldsymbol{b}_k}$$

$$= \frac{1}{n_{\text{data}}} a_i (\boldsymbol{s}_k)^\top (\rho_1 a_1 \boldsymbol{t}_1 + \ldots + \rho_N a_N \boldsymbol{t}_N)$$

$$+ \frac{1}{n_{\text{data}}} a_k \boldsymbol{t}_k^\top \rho_k a_k \boldsymbol{s}_k.$$

Combining the above, we obtain $\frac{\partial \boldsymbol{u}}{\partial \boldsymbol{b}}$, and hence (49) follows.

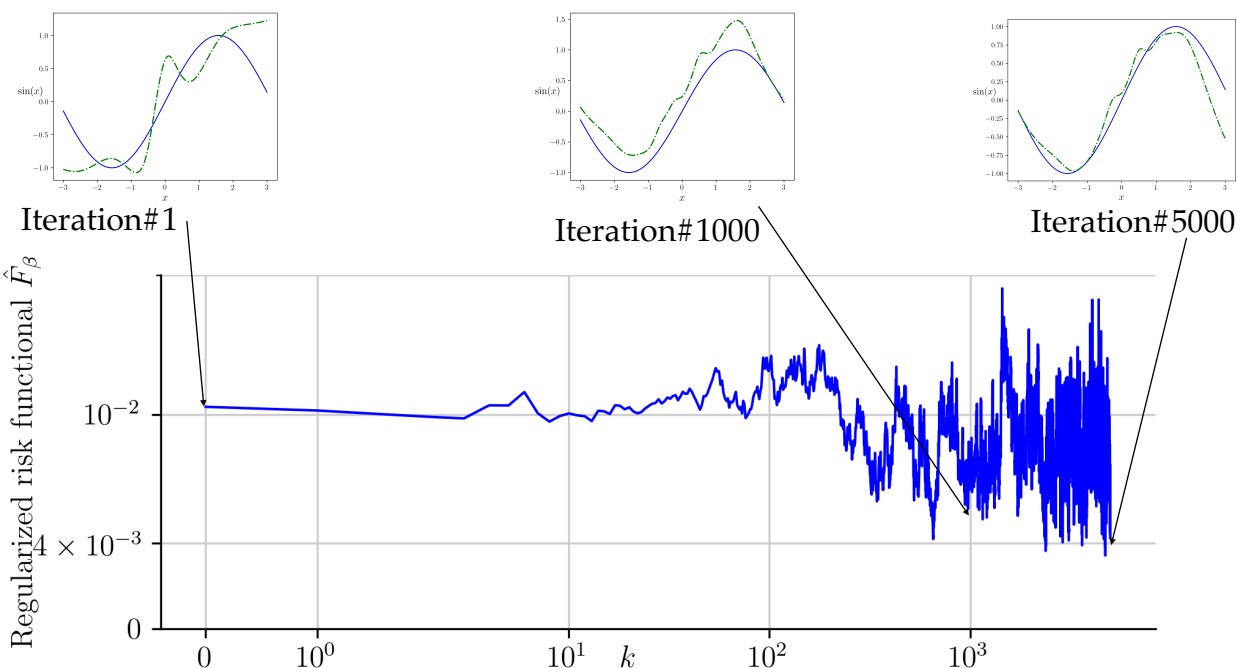

Figure 4: The evolution of the regularized risk $\hat{F}_\beta$ versus iteration index $k$ for the proposed PROXLEARN. Inset plots compare the ground truth (sinusoid) with the output from the network at three specific iterations.

Finally, noting that $\boldsymbol{W}_{i,j}$ is in the $i$th row of $\boldsymbol{T}$, for $i \neq k$, we obtain

$$\frac{\partial \boldsymbol{u}_k}{\partial \boldsymbol{W}_{i,j}} = \frac{1}{n_{\text{data}}} a_k \boldsymbol{t}_k^\top \rho_i a_i \frac{\partial \boldsymbol{t}_i}{\partial \boldsymbol{W}_{i,j}} = \frac{1}{n_{\text{data}}} a_k \boldsymbol{t}_k^\top \rho_i a_i (\boldsymbol{s}_i \odot \boldsymbol{x}_j),$$

where $\boldsymbol{x}_j$ is the $j$th column of $\boldsymbol{X}$. Likewise, for $i = k$, we get

$$\begin{aligned}
\frac{\partial \boldsymbol{u}_k}{\partial \boldsymbol{W}_{i,j}} &= \frac{1}{n_{\text{data}}} a_k \left( \frac{\partial \boldsymbol{t}_k}{\partial \boldsymbol{W}_{i,j}} \right)^\top (\rho_1 a_1 \boldsymbol{t}_1 + \ldots + \rho_N a_N \boldsymbol{t}_N) \\
&\qquad + \frac{1}{n_{\text{data}}} a_k \boldsymbol{t}_k^\top \rho_k a_k \frac{\partial \boldsymbol{t}_k}{\partial \boldsymbol{W}_{i,j}} \\
&= \frac{1}{n_{\text{data}}} a_k (\boldsymbol{s}_i \odot \boldsymbol{x}_j)^\top (\rho_1 a_1 \boldsymbol{t}_1 + \ldots + \rho_N a_N \boldsymbol{t}_N) \\
&\qquad + \frac{1}{n_{\text{data}}} a_k \boldsymbol{t}_k^\top \rho_k a_k (\boldsymbol{s}_i \odot \boldsymbol{x}_j).
\end{aligned}$$

Combining the above, we obtain $\frac{\partial \boldsymbol{u}}{\partial \boldsymbol{W}_{i,j}}$, thereby arriving at (50). □

## D  Learning sinusoid

To better visualize the functionality of the proposed algorithm, we perform a synthetic case study of learning a sinusoid following the set up as in (Novak et al., 2019, Sec. 2.1). We performed 5000 iterations of PROXLEARN with $N = 1000$ samples (no mini-batch) from the initial PDF $\varrho_0(\boldsymbol{\theta} \equiv (a, b, w)) = \text{Unif}([-1, 1] \times [-1, 1] \times [-1.5, 1.5])$, and used algorithm parameters $\beta = 0.3$, $h = 10^{-4}$, $\delta = 10^{-3}$, $\varepsilon = 10^{-3}$, $L = 10$. The evolution of the associated regularized risk functional and the learnt functions are shown in Fig. 4. Fig. 5 shows the function approximations learnt by our proposed algorithm at the end of 3200 iterations for 20 randomized runs with the same initial PDF and parameters as reported here.

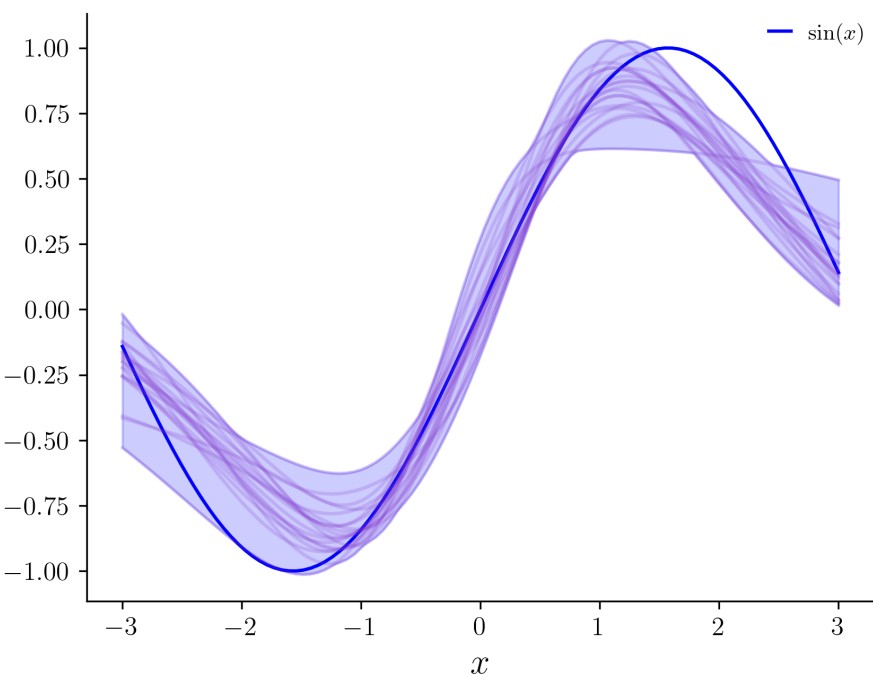

Figure 5: Comparison of the ground truth (here $\sin(x)$) with the learnt approximants obtained from the proposed PROXLEARN after 3200 iterations for 20 randomized runs. All randomized runs use the same initial PDF and parameters as reported here.

Table 5: Comparing final $\hat{F}_\beta$ for varying $N$

| $N$ | Final $\hat{F}_\beta$ |
|------|------------|
| 500 | 0.01241931 |
| 700 | 0.01075817 |
| 1000 | 0.00806645 |
| 2000 | 0.00762518 |

To illustrate the effect of finite $N$ on the algorithm's performance, we report the effect of varying $N$ on the final regularized risk value $\hat{F}_\beta$ for a specific synthetic experiment. We observe that increasing $N$ improves the final regularized risk, as expected intuitively.

