# OpenReview forum: "Proximal Mean Field Learning in Shallow Neural Networks"
_TMLR — Accepted by TMLR_

### Review · Reviewer_7qYb · 2023-09-15

**Summary Of Contributions:**

This paper introduces a proximal algorithm for a Wasserstein gradient flow in the space of probability measures to carry out shallow neural network training in the mean-field regime. The main contributions are the construction of an algorithm to carry out this scheme in practice (Eq. 34), a convergence result ensuring asymptotic convergence to a unique fixed point, and numerical experiments assessing the performance of the method.

**Audience:**

Yes

**Claims And Evidence:**

Yes

**Requested Changes:**

I think the authors should study the role of $\epsilon$ and give better assurances of stability in their implementation. I would also like some analysis of the time evolution of the risk---why does it seem to suddenly start decreasing only after $10^5$ iterations?

**Strengths And Weaknesses:**

The paper is very clearly written and does an admirable job of discussing related work. The motivation for and construction of the algorithm is done very precisely and approximations are mentioned throughout. Comparing the limiting dynamics to particle-based methods remains a significant topic of interest, so I believe this numerical approach is also timely and important.

For me, the main weaknesses involve the numerical results: first, the role of regularization parameter in the Sinkhorn is not really described in terms of either the stability or accuracy of the results. It would be good to include some numerical results that show how performance varies with the degree of regularization.

Second, in both the weighted and unweighted cases, the evolution of the loss is not particular reassuring (Fig. 2). The risk functional does not decay smoothly (as opposed to Fig. 3) and the numerical stability looks rather questionable.

---

> ### Author Response · Authors · 2023-11-16
> **On the numerics w.r.t. $\varepsilon$ and the decay of regularized risk**
>
> We appreciate your perceptive comments and feedback to help us improve the manuscript.
>
> **Response to the weakness/requested change on numerical stability w.r.t. regularization parameter $\varepsilon$**
>
> Following the reviewer's suggestion, in the revised manuscript, we have included the new Table 3 reporting additional experiments that shows variations in final regularized risk w.r.t. variations in $\varepsilon$. As intuitively expected, larger $\varepsilon$ entails more smoothing and lowers runtime (last column in Table 3). The corresponding final regularized risk values reported in the middle column of  Table 3 show no significant variations, suggesting numerical stability. This new Table 3 is discussed inline in a new paragraph included in the revised version at the end of Sec. 4.2.
>
> **Response to the weakness/requested change on time evolution of the risk---why does it seem to suddenly start decreasing only after $10^5$ iterations?**
>
> In the revised manuscript's Sec. 4.4 beginning, we have added a new paragraph with references discussing the long flat region in the evolution of the risk, i.e., significant burn in period that the reviewer noted. In that paragraph, we point out that the same trend (speed up only after  $10^{5}$ iterations) was observed and explicitly stated in other works that considered the same mean field limiting PDE initial value problem. We also discuss why this trend is possibly a characteristic of the particular mean field limit PDE as opposed to the algorithms used.

---

### Review · Reviewer_QGHP · 2023-10-22

**Summary Of Contributions:**

This paper proposes a proximal algorithm to approximate the distributional flow of the learning dynamics of shallow neural networks. Such method tolerates evolving the neuronal population distribution in an online manner. Also, the motivation and derivation of the algorithm is supported by solid mathematical works.

**Audience:**

Yes

**Broader Impact Concerns:**

No such concerns.

**Claims And Evidence:**

Yes

**Requested Changes:**

1. Sections 2 and 3 are too detailed with many equations. I don't think they are needed in the main body of the paper. It makes it difficult to follow the paper. Some major insights and conclusions should be emphasized to make the contribution clearer. Some discussions and explanations, e.g., for Theorem 1, sections 3.1 and 3.2,  can be added to help the understanding.

2. For experiments, (a) Section 4.3 seems irrelevant to your algorithm. If so, it is better to remove it. (b) I think Table 3 shows that the results are worse than existing methods. (c) it is better to add the comparison of computational complexities between different methods.

**Strengths And Weaknesses:**

Strengths:
1. The logic of this paper is clear.
2. The derivation seems solid and correct.

Weaknesses:
1. Writing needs some improvement.
2. Some parts of the experiments are confusing.

---

> ### Author Response · Authors · 2023-11-17
> **On writing improvements**
>
> We thank the reviewer for the kind feedback to help us improve.
>
> **Response to weaknesses**
> - Writing improvement: Please see our responses below to the requested change #1. Furthermore, we added in new discussions at the end of Sec. 1.2 citing literature where other types of mean field learning are considered. That paragraph contextualizes this work w.r.t. broader mean field learning literature, which should help the readers better appreciate/contrast our contribution.
>
> - Confusion on some parts of the experiments: Please our responses to the requested change #2 below where we made an effort to clarify.
>
> **Response to requested changes**
>
> #1. As suggested, in the revised manuscript, we have added an informal statement for Theorem 1 in the main body. Its rigorous statement and the proof are deferred to Appendix A. Furthermore, to help the understanding, the revised version's Sec. 3.1 now includes an opening paragraph outlining the main insight behind the derivation that follows. We decided against further reducing Sections 2 and 3 because doing so makes it very difficult to clearly explain the notations and technical ideas that follow.
>
> #2. (a) As per the reviewer's suggestion, in the revised version, we have de-emphasized the the previous version's Sec. 4.3 by removing the corresponding subsection heading and making it instead part of the preceding text detailing numerical experiments.
>
> (b) From the revised version's Table 4, third column, the proposed "ProxLearn, Weighted" is shown to achieve comparable accuracy (difference in second or third decimal places) to the two existing state-of-the-art (JKO-ICNN, SWGF + RealNVP) under comparison for the same datasets. Please note that only the "ProxLearn, Weighted" use the the ensemble average using the proximal updates; see: Sec. 4.1, first paragraph's last sentence. Please also see the revised version's Sec. 4.4, second paragraph, first sentence, and the last paragraph of Sec. 1.3 -- in both places we clearly state that the purpose and the perspective of this work is not to beat the carefully engineered state-of-the-art right way but to show that the mean field learning can indeed lead to a feasible algorithm. Currently, this idea is a theoretical tool and we discuss why making it an algorithm is not quite straightforward. Also note that we did not optimize any hyperparameters to achieve the reported simulation performance, since that is not the objective of this work. They are reported as is.
>
> (c) Thanks for this suggestion. In the revised version's Sec. 4.3, last paragraph, where we mention the complexity of the proposed method, we have added in the computational complexities for JKO-ICNN and SWGF too.

---

> > ### Comment · Reviewer_QGHP · 2023-11-25
> >
> > Thank you for the response. My major concerns are addressed. The authors did a great job in revising the paper.

---

### Review · Reviewer_kope · 2023-11-02

**Summary Of Contributions:**

This manuscript proposes a novel optimization algorithm for shallow neural networks.  In particular, this looks at a PDE approach to neural network learning and derives a new algorithm to approximate the neural network solution by weighted scattered particles.  The derivation of this algorithm requires significant mathematical developments.

**Audience:**

Yes

**Broader Impact Concerns:**

None.

**Claims And Evidence:**

Yes

**Requested Changes:**

I have three required changes:

1. A bigger discussion about the potential long-term utility of this work, including more specific examples of how this could be used either theoretically or applied going forward.
2. A comparison of how different functions learned by this approach versus other approaches are.
3. A discussion of the learning dynamics, and why the algorithm is so slow in early iterations.

Some additional suggested changes:
1. There's not much evidence on what the different parameters of the algorithm do.  While the primary goal is a proof-of-concept, it would be beneficial to understand what the tuning parameters do.  I'm particularly interested in $N$.
2. Algorithm 1 and 2 are challenging to read.  I'd suggest adding comments throughout the algorithms to explicitly describe the steps and link them back to the paper.  Furthermore, it may be useful to further break up Algorithm 1 for clarity.

**Strengths And Weaknesses:**

## Strengths

There is quite a lot of seemingly novel math here.  The overall logic of the algorithm is easy to follow despite the complexity caused by the many mathematical steps and operations.

This provides a different viewpoint on learning in shallow neural networks.  This viewpoint is potentially valuable to really understand the behavior of how learning happens in greater detail.

## Weaknesses

In my opinion, the greatest weakness of this work is that it is not clear how the developments in this paper can be used and built upon.  The author's highlight at the end of Section 1.3
> This is a new line of idea for learning algorithm design that we show is feasible. As such, we do not aim to immediately surpass the carefully engineered existing state-of-the-art in experiments. Instead, this study demonstrates a proof-of- concept which should inspire follow up works.

I generally think that it's good to explore new ideas, but the claim here is rather vague.  What are the follow-up works?  How could this be used?  I don't see how a similar framework would be created for deeper models.  This manuscript would be greatly enhanced by discussing at much greater length how the framework could be used in future work.

The overall landscape of analysis of shallow networks should be enhanced.  For example, there are the viewpoints where we can view and learn with kernels or gaussian processes.

I would really like to understand what this algorithm is learning functionally and how it compares to alternative approaches.  I would suggest looking at synthetic data with 1-dimensional inputs so that you can visualize the input-output space, and then looking at the functions learned by this approach versus others.  How does it compare to gradient descent or kernel approximations functionally.  For example, consider Figure 1 in [1].

[1] Novak, Roman, et al. "Neural tangents: Fast and easy infinite neural networks in python." ICLR 2020.

## Questions

The learning curves in Figure 2 are unusual.  Why is there such a significant burn-in period where the risk functional does not get better?  Why is the discrepancy so large between the GPU and CPU solutions? e.g., the GPU has big jumps where the risk functional gets worse.  I understand that they have different numerical precisions, but I wouldn't expect them to be so different.

---

> ### Author Response · Authors · 2023-11-17
> **Thank you for the constructive feedback**
>
> We thank the reviewer for careful reading and for providing constructive feedback to help us improve the manuscript. Please find our itemized responses below.
>
> **Response to Questions**
>
> Q. "The learning curves in Figure 2 are unusual. Why is there such a significant burn-in period where the risk functional does not get better?"
>
> A. The reviewer is right that there is a significant burn in period. In fact, these trends in learning curves agree with those reported in (Mei et. al. 2018, PNAS) which we cited in our manuscript. In particular, Fig. 3 in (Mei et. al. 2018, PNAS) and Fig. 7.3 in that paper's *Supporting Information*, show convergence trends very similar to our Fig. 2: slow decay until approx. $10^{5}$ iterations and then a significant speed up. The unusual convergence trend was also stated in text in (Mei et. al. 2018, PNAS): ``We  observe  that  SGD converges to a network with very small risk, but this convergence has a nontrivial structure and presents long flat regions." It is interesting to note that (Mei et. al. 2018, PNAS) considered an experiment that allowed rotational symmetry and simulated the radial (i.e., one spatial dimensional) discretized PDE, while we used the proposed proximal recursion directly in the neuron population ensemble to solve the PDE IVP, i.e., similar convergence trends were observed using different numerical methods applied to the same mean field PDE IVP. This makes us speculate that this convergence trend is specific to the mean field PDE dynamics itself, and is less about the specific numerical algorithm. Recent works such as (Wojtowytsch and Weinan, 2020) investigate mean-field learning dynamics in two-layer ReLU networks and in that setting, show that the learning can indeed be slow depending on the target function.
>
> Q. Why is the discrepancy so large between the GPU and CPU solutions? e.g., the GPU has big jumps where the risk functional gets worse. I understand that they have different numerical precisions, but I wouldn't expect them to be so different.
>
> A. We agree that the differing choices for numerical precision places a significant role in creating these apparent discrepancies. Additionally, please note that the numerical experiment conducted via GPU ran for fewer iterations than the numerical experiment calculated via CPU ($2.5 \times 10^{5}$ vs $10^6$). The result is different scaling in the two figures, which also contributes to the different solutions. When conducting the GPU experiments, our goal was to minimize computational time while maintaining equivalent classification accuracy; hence the reduction in total iterations.
>
> **Response to requested changes**
>
> *Required changes*
>
> Q1. A bigger discussion about the potential long-term utility of this work, including more specific examples of how this could be used either theoretically or applied going forward.
>
> A1.  In the revised manuscript, we have re-structured the Sec. 6 (Conclusions) in two subsections. The current Sec. 6.1 contains the summary of the present contribution. The material of Sec. 6.2 is new--it specifically discusses how and why the proposed ideas can be pursued for multi-layer networks going forward. Along the way, we explain the different mean field limits for multi-layer case and why one of them is more suitable for our framework.
>
> Q2. A comparison of how different functions learned by this approach versus other approaches are.
>
> A2. In the revised manuscript’s Sec. 1.2, we have added a paragraph with new references comparing different viewpoints on mean field learning including the NTK and gaussian processes, and contrasting the same with the present work. Motivated by the Fig. 1 in ref. [1] that the reviewer suggested, we also performed the 1D case study of learning $\sin(x)$ following the same set up as in the Sec. 2.1 of that reference using our proximal algorithm. We report this numerical simulation in the new Appendix D. We slightly expanded the end of Sec. 1.5 to refer to the same.
>
> Q3. A discussion of the learning dynamics, and why the algorithm is so slow in early iterations.
>
> A3. Following the reviewer's suggestion, in the revised manuscript's Sec. 4.4 beginning, we have added a new paragraph with references discussing the long flat region, i.e., significant burn in period that the reviewer noted. We point out that the same trend was observed and explicitly stated in other works and speculate why this trend is possibly a characteristic of the particular mean field limit PDE as opposed to the algorithms used.
>
> *Suggested changes*
>
> A1. To clarify numerical stability w.r.t. the regularization $\varepsilon$, we have added new Table 3 reporting additional numerical experiments, and discussed the same in a new paragraph at the end of Sec. 4.2.  The effect of varying $N$ is reported in the new Appendix D.
>
> A2. As per the reviewer's suggestion, in the revised manuscript, we have added comments in Algorithms 1 and 2, including references to the derivation of the relevant equations.

---

### Author Response · Authors · 2023-11-17
**We thank the reviewers for helpful comments**

We are indebted to the reviewers for their helpful comments and constructive feedback which have helped us improve the manuscript. We have provided itemized responses accounting the reviewers' suggestions, and detailing the corresponding changes in the revised version.

---

### Decision · Action_Editor_7DGr · 2023-12-11

**Recommendation:** Accept as is

**Comment:**

Given the above 2 points, I am happy to recommend this paper for acceptance into TMLR. The authors have already made several updates to their paper in light of the reviews, and the paper can be accepted as-is (after changing blue changes back to black).

**Audience:**

At least some individuals in TMLR's audience will be interested in the findings of this paper. All reviewers agree.

The authors consider mean-field training of a shallow infinitely wide (overparameterised) neural network. The infinite width model is used as a computational driver rather than a conceptual analysis framework. They design a Sinkhorn regularized proximal algorithm to approximate the distributional flow for the learning dynamics over weighted point clouds. While such models are not expected to be state of the art in many settings, there is definitely a wide ML audience who will be interested in the findings of this paper.

**Claims And Evidence:**

Claims in the submission are supported by accurate, convincing and clear evidence. All reviewers agree.

Reviewer 7qWb initially pointed out that "the role of regularization parameter in the Sinkhorn is not really described in terms of either the stability or accuracy of the results", and this has been adequately addressed by the authors in their updated manuscript. Each reviewer pointed out minor presentation and clarity issues, and this has helped improve the quality of the paper.

---

> ### Author Response · Authors · 2023-12-14
> **Thank you for the decision and review process**
>
> We thank the editors and all the reviewers for their perceptive inputs helping improve the quality of the manuscript. The efficient review process is much appreciated.
>
> Sincerely,
> Authors